# Harnessing natural variation to identify *cis* regulators of sex-biased gene expression in a multi-strain mouse liver model

**Bryan J. Matthews**[1], **Tisha Melia**[1,2], **David J. Waxman**[1,2]*

**1** Department of Biology, Boston University, Boston, Massachusetts, United States of America,
**2** Bioinformatics Program, Boston University, Boston, Massachusetts, United States of America

* djw@bu.edu

## Abstract

Sex differences in gene expression are widespread in the liver, where many autosomal factors act in tandem with growth hormone signaling to regulate individual variability of sex differences in liver metabolism and disease. Here, we compare hepatic transcriptomic and epigenetic profiles of mouse strains C57BL/6J and CAST/EiJ, representing two subspecies separated by 0.5–1 million years of evolution, to elucidate the actions of genetic factors regulating liver sex differences. We identify 144 protein coding genes and 78 lncRNAs showing strain-conserved sex bias; many have gene ontologies relevant to liver function, are more highly liver-specific and show greater sex bias, and are more proximally regulated than genes whose sex bias is strain-dependent. The strain-conserved genes include key growth hormone-dependent transcriptional regulators of liver sex bias; however, three other transcription factors, *Trim24*, *Tox*, and *Zfp809*, lose their sex-biased expression in CAST/EiJ mouse liver. To elucidate the observed strain specificities in expression, we characterized the strain-dependence of sex-biased chromatin opening and enhancer marks at *cis* regulatory elements (CREs) within expression quantitative trait loci (eQTL) regulating liver sex-biased genes. Strikingly, 208 of 286 eQTLs with strain-specific, sex-differential effects on expression were associated with a complete gain, loss, or reversal of the sex differences in expression between strains. Moreover, 166 of the 286 eQTLs were linked to the strain-dependent gain or loss of localized sex-biased CREs. Remarkably, a subset of these CREs apparently lacked strain-specific genetic variants yet showed coordinated, strain-dependent sex-biased epigenetic regulation. Thus, we directly link hundreds of strain-specific genetic variants to the high variability in CRE activity and expression of sex-biased genes and uncover underlying genetically-determined epigenetic states controlling liver sex bias in genetically diverse mouse populations.

## Author summary

Male-female differences in liver gene expression confer sex differences in diverse biological processes relevant to human health and disease but are difficult to model in inbred

---

**Data Availability Statement:** Data availability RNA-seq data for male and female B6 and CAST mouse liver are available under accession number GSE130913 (https://www.ncbi.nlm.nih.gov/geo/

---

**Funding:** This work was supported in part by NIH grant DK121998 (to DJW). The funders had no role in study design, data collection and analysis, decision to publish, or preparation of the manuscript.

mice given their identical genetic backgrounds. Outbred mice provide some variability, but cross-strain studies of sex bias in rodents have not been well studied. Here we elucidate the actions of genetic factors regulating liver sex differences in two Diversity Outbred mouse founder mouse strains, C57BL/6 and CAST/EiJ. We find that many of the strain differences in sex-biased gene expression can be linked to the gain or loss of a *cis* regulatory element associated with one or more strain-specific sequence variants. Strikingly, in many cases, the associated *cis* regulatory element lacked strain-specific variants, yet was subject to coordinated, strain-dependent epigenetic regulation. Thus, harnessing the power of naturally occurring genetic diversity of Diversity Outbred mice, we integrated biological data at the genetic, epigenetic, and transcriptomic levels across evolutionary divergent mouse strains to discover hundreds of localized genomic regions that control phenotypic sex differences in the liver. These findings may serve as a model for studies of human genetic variation and the effect of population-wide variation on sex differences in health and disease.

## Introduction

Many vertebrate tissues, including non-reproductive tissues, show significant sex differences in their gene expression profiles, metabolic and physiological properties, and patterns of disease susceptibility [1–3]. Underlying regulatory mechanisms are best studied in the liver, where there is extensive transcriptomic and regulatory sex bias in fish [4,5], rats [6,7], mice [8–10] and humans [11,12]. In mouse liver, hundreds of genes are expressed in a sex-dependent manner, including protein-coding genes [13], miRNAs [14,15] and lncRNA genes [16–18]. Phenotypically, these sex differences in expression contribute to sex differences in chemical sensing and metabolism [19–21], response to injury [22,23], and susceptibility to disease [24–27].

Growth hormone (GH) has been directly implicated as the major proximal regulator of sexual dimorphism in the liver. Pituitary GH secretion patterns, which are controlled by the hypothalamus in a sex-dependent manner [28,29], activate GH signaling to the nucleus through the JAK/STAT pathway [30,31] and are crucial to hepatic sex differences in both rodents and humans [20,32–34]. Thus, sex-biased gene expression in the liver is substantially lost when GH secretion is ablated (>90% of sex-biased genes) [35,36] or following continuous GH infusion, which overrides the endogenous male, pulsatile plasma GH profile (74% of sex-biased genes) [13]. Major dysregulation of hepatic sex-biased genes also occurs upon loss of the GH receptor-activated proximal transcriptional regulator STAT5 [9,37,38] or its downstream targets, notably the transcriptional repressors Bcl6 [39,40] and Cux2 [41].

Genome evolution results in species differences in transcription factor binding at *cis* regulatory elements (CREs), with a strong preference for evolution of regulatory sequences at enhancers as compared to gene promoters [42,43]. Analysis of such evolutionary changes at CREs between closely related species, or across strains within a species, provides an opportunity to discover functional roles of specific regulatory elements [44–46]. Genetic diversity across mouse strains and subspecies results in well-characterized phenotypic [47,48], transcriptomic [49–51], and epigenetic differences in the liver [44–46]. Furthermore, extensive transcriptomic and epigenetic studies on liver sex differences in the outbred mouse strain CD-1 have identified large numbers of GH-regulated CREs that differ between the sexes [38,52–54]. The sex bias of hepatic gene expression has also been described in several inbred mouse strains [8,10,55]. Underlying mechanisms for genetically-based differences between strains are

poorly understood [49,56–59], primarily because most studies of sex bias in mouse liver have focused on strains of primarily *M. m. domesticus* genetic background (i.e., CD-1 and C57BL/6J) and did not include wild-derived strains.

The Diversity Outbred (DO) mouse model, where known genetic variants from eight well-characterized inbred founder mouse strains have been reshuffled through a specific multi-generation breeding program [60,61], is an important resource for studying the contributions of genetic variation to diversity. The genetic diversity of individual DO mice [62] is comparable to the diversity of the global human population [63] and serves as a relevant model for diversity in gene expression and disease in humans. Livers from individual DO mice differ in their responses to chemical exposure [64] and western diet [65], and show marked differences in expression of sex-biased genes [17]. Furthermore, recent work from this laboratory analyzing RNA-seq datasets from 438 individual DO mice identified 1,137 expression quantitative trait loci (eQTLs), each harboring genetic variants with a significant impact on the expression of sex-biased genes [49]. However, the loci identified are quite large (median width = 1.84 Mb) and contain large numbers of strain-specific SNPs and Indels, making it difficult to link specific genetic variants to the variable expression of individual sex-biased genes.

Here, we use a combination of DNase-seq, ChIP-seq and RNA-seq to capture sex differences, and their strain dependence, at the epigenetic and transcriptomic levels for two DO mouse founder strains, C57BL/6J (B6) and CAST/EiJ (CAST) mice. These strains—B6, an inbred laboratory mouse strain composed of primarily subspecies *Mus musculus domesticus* genetic background, and the wild-derived CAST strain, of subspecies *Mus musculus castaneus* —comprise two of the eight founder strains originally used to generate DO mice [60,61] and are separated by 0.5–1 million years of evolution [66,67]. We identify > 200 strain-conserved sex-biased protein coding and lncRNA genes between B6 and CAST mouse liver, including several core transcription factors regulating liver sex differences, which likely underpin the conservation of phenotypic sex bias across strains. Sex-biased expression was lost in CAST liver for three B6 sex-biased transcriptional regulators, which may contribute to the strain-dependence of sex-biased gene expression through a *trans* effect. Further, we identify hundreds of strain-conserved, sex-biased CREs, 80% of which have properties of gene-distal enhancers, and we elucidate the strain-specific gain or loss of many hundreds of other sex-biased CREs associated with genetic variation within eQTL regions. Importantly, a majority of the strain differences that we evaluated in the set of eQTLs regulating strain-specific, sex-biased gene expression were directly associated with a gain or loss of CREs, but remarkably, a subset of these CREs were devoid of strain-specific SNPs/Indels. Thus, by integrating biological data at the genetic, epigenetic, and transcriptomic levels across evolutionarily divergent inbred mouse strains, we identify genomic sequences likely to serve as causal regulators of the transcriptional activity of a significant subset of the sex-biased murine liver transcriptome.

## Results

### Strain-shared and strain-unique sex-biased genes in B6 and CAST mouse liver

We performed RNA-seq to identify genes showing sex-biased expression in B6 and CAST mouse livers. Results were compared to prior data for sex-biased genes in livers of the outbred mouse strain CD-1 [13], which is most similar to the DO founder mouse strain NOD/ShiLt and is particularly useful due to its inherent genetic variability [68]. 946 protein-coding genes and 751 lncRNA genes showed significant liver sex-bias in one or more strains (Fig 1A and S1 Table). We observed a greater overlap of sex-biased genes between CD-1 and B6 relative to CAST (229 protein coding genes versus 144, respectively; Fig 1A), which is likely because both

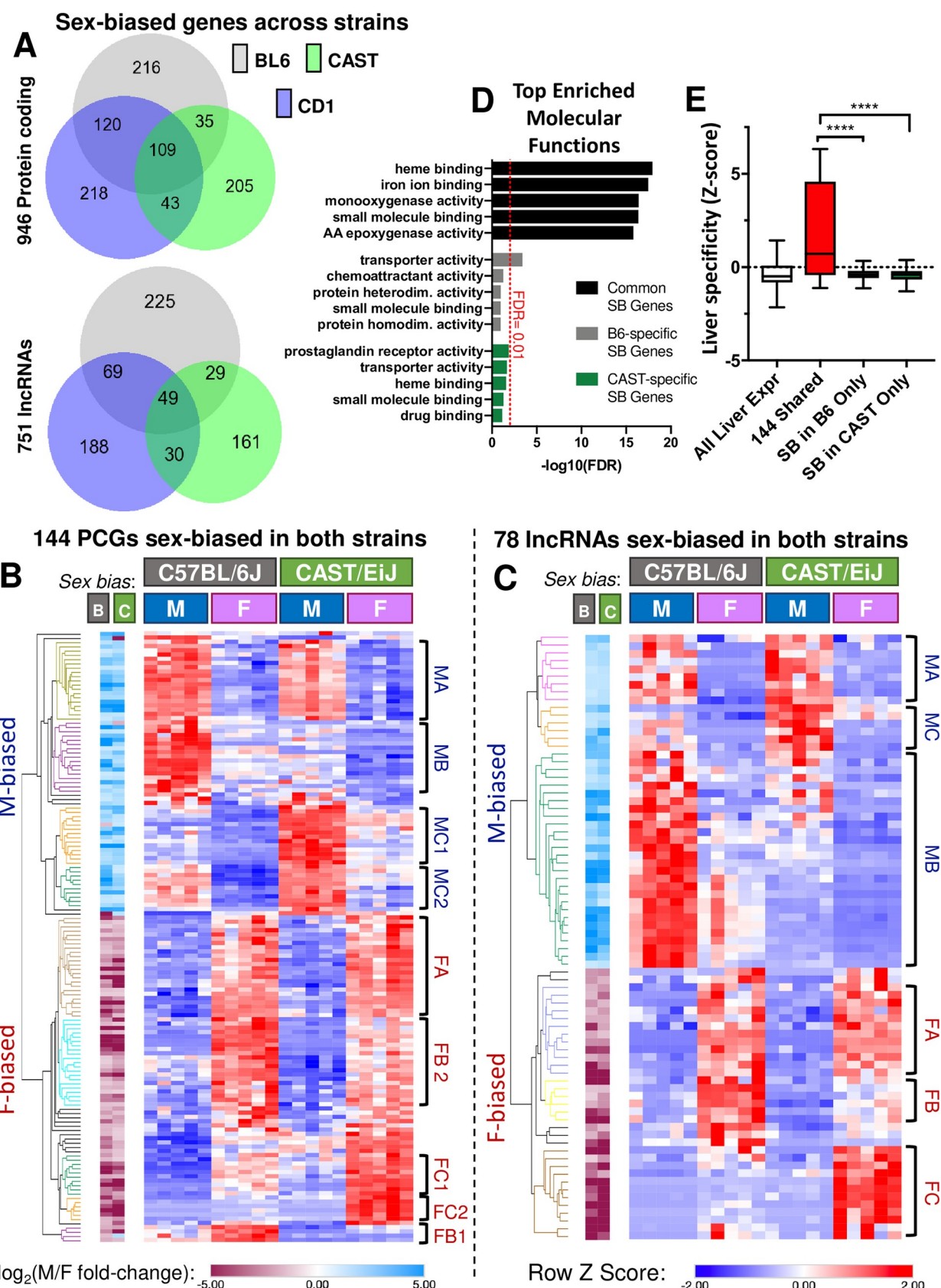

**A** Sex-biased genes across strains

BL6  CAST  CD1

946 Protein coding

751 lncRNAs

**D** Top Enriched Molecular Functions

Common SB Genes
B6-specific SB Genes
CAST-specific SB Genes

FDR = 0.01

-log10(FDR)

**E**

**B** 144 PCGs sex-biased in both strains

**C** 78 lncRNAs sex-biased in both strains

log₂(M/F fold-change):

Row Z Score:

**Fig 1. Sex-biased transcriptome of B6 and CAST mouse liver. A**. Venn diagrams showing the number of strain-shared and strain-unique sex-biased protein-coding genes (*top*) and lncRNA genes (*bottom*) for the three indicated mouse strains, based on |fold change| for sex bias >2 and EdgeR adjusted p-value (FDR) < 0.05. CAST (green) and B6 (gray) RNA-seq datasets are from this study; CD-1 RNA-seq data is from [13]. B6 shows greater overlap with CD-1 than with CAST. **B**. Heat map presenting relative expression level across all individual livers (n = 20) for 144 protein-coding genes showing significant sex bias in both B6 and CAST mice (|fold-change| > 2, FDR < 0.05; strict strain shared sex-biased genes, Sheet A in S1 Table), presented as Z-scores normalized per row to visualize expression patterns independent of expression level. *Left*: Log2 (M/F fold-change) values; blue indicates male bias, purple indicates female bias for B6 (B, *left column*) and CAST (C, *right column*). Hierarchical clustering based on Euclidean distance is shown to the left of the heat map, with colors indicating the cluster identity labelled to the right of the heat map. Two of the 9 clusters showed a strain-conserved pattern of expression between B6 and CAST (clusters MA and FA). The other 7 clusters were significantly sex-biased in expression in both strains but showed strain-variable levels of expression (CAST-biased: clusters MC1, MC2, FC1, and FC2; B6-biased: MB, FB1, and FB2). Clusters labeling: M or F indicate the sex-bias, and B (B6) or C (CAST) indicate the strain with higher expression; A indicates clusters with strain-equivalent expression. **C**. Heat map, as in B, for 78 sex-biased lncRNA genes meeting fold-change and FDR cutoffs in both B6 and CAST mice (strict strain shared sex-biased genes; Sheet A in S1 Table). Two of the six clusters showed a strain-conserved pattern of expression between B6 and CAST (clusters MA and FA). The other four clusters were significantly sex-biased in both strains but showed strain-variable levels of expression (CAST-biased: clusters MC and FC; B6-biased: MB and FB). **D**. Genes with sex bias in both B6 and CAST mouse liver are enriched for key liver functions vs. no GO term enrichment for genes with sex bias only in one strain. Shown is -log$_{10}$(FDR) from functional annotation of the indicated gene lists output by DAVID for the category: GOTERM_MF_DIRECT. Black bars, results for core, strain-conserved sex-biased genes; gray bars, B6-specific sex-biased genes; green bars, CAST-specific sex-biased genes. Shown are the top 5 significant GO terms ranked by FDR. Red line, FDR cutoff at 0.01. See Sheets A-C in S2 Table for complete lists of terms. **E**. Per-gene Z score distribution for expression across 21 ENCODE mouse tissues [69] implemented as described [74]. Y-axis values indicate increasing liver specificity relative to the mean expression across tissues. Z scores are shown (from left to right) for: all liver-expressed genes, 144 core sex-biased genes, 301 B6-unique sex-biased genes, and 207 CAST-unique sex-biased genes. ****, M-W p < 0.0001 for 144 core, strain-conserved sex-biased gene set versus both strain-specific gene lists. The strain-specific sex-biased genes do not differ significantly (M-W p = 0.48).

strains are composed of primarily subspecies *Mus musculus domesticus* genetic background [66–68]. Comparing the two evolutionarily distant inbred strains, B6 and CAST, we identified 144 protein-coding genes and 78 lncRNA genes with strain-shared sex bias (defined as significant sex bias in both B6 and CAST). These strain-shared sex-biased genes were clustered and annotated based on their relative expression across individual livers from each strain and sex (Fig 1B). Importantly, the directionality of sex bias was conserved between strains for almost all the strain-shared sex biased genes (217 of 222 genes; Fig 1B and 1C; examples in S1 Fig and full gene lists in Sheets A and B in S1 Table). However, only 29% of the core strain-shared sex-biased genes (43/144 protein-coding genes, 21/78 lncRNAs) showed the same expression pattern in livers of both strains (clusters MA and FA; Fig 1B and 1C). The remaining 71% of strain-shared genes showed strain differences in the absolute expression level (FPKM) and magnitude of sex bias (e.g., genes in clusters MB, FB1, and FB2 showed higher expression in B6 than CAST). Accordingly, while most of the variation in expression of strain-shared sex-biased genes is associated with differences in sex (PC1, 60–66% of variance), substantial variation is explained by strain differences (PC2, 22–26% of variance; S1E and S1F Fig). Finally, we identified genes whose sex-biased expression in liver is specific to one strain. Thus, B6-unique: 301 protein-coding genes, 289 lncRNAs; and CAST-unique: 207 protein-coding genes, 187 lncRNAs. A majority of these strain-unique sex-biased genes are more highly expressed in the strain showing significant sex-bias (S2 and S3 Figs and Sheets C-F in S1 Table).

## Functional roles of strain-shared vs strain-unique sex-biased genes

Despite substantial evolutionary distance between B6 and CAST mice, sex-biased expression is maintained for a core set of functionally relevant and liver-specific genes. The core set of 144 strain-shared sex-biased protein-coding genes (Fig 1B) includes many *Cyp*, *Mup*, and *Sult* genes, and showed strong enrichment for molecular function gene ontologies relevant to liver function, including monooxygenase activity and small molecule binding/lipocalin (Fig 1D). In contrast, only one Gene Ontology term (transporter activity) was significantly enriched in the set of 301 B6-unique sex-biased protein-coding genes, and no molecular function terms were enriched in the 207 CAST-specific gene set (FDR<0.01; S2 Table). Furthermore, the strain-

shared sex-biased genes are expressed in a highly liver-specific manner, as compared to either the strain-unique gene sets or all liver expressed genes, based on comparisons to 21 mouse ENCODE tissues [69] (Mann-Whitney (M-W) p < 0.001; Fig 1E).

The major GH-regulated or GH-responsive transcription factors (TFs) implicated in liver sex differences in CD-1 mice were expressed at similar levels in both mouse strains, consistent with a model of evolutionary conservation of functionally relevant genes. These include the GH-responsive TFs STAT5b [9], Foxa2 [70] and Hnf4a [37], and the GH-regulated, sex-biased TFs Cux2 [71], Bcl6 [40] and Sall1 [13] (S4A and S4B Fig and Sheet A in S3 Table). The sex-biased expression of five other TF genes was strain-dependent (Sheets B and C in S3 Table); these include three B6-unique sex-biased TFs previously identified as being GH-responsive (Trim24, Tox, and Zfp809; S4C Fig) [13,71,72] and two CAST-unique sex-biased TFs with no known roles in liver sex differences (Id3, Rnf141; S4D Fig). Conceivably, the three B6 mouse-specific TFs may contribute to the larger number of sex-biased genes found in B6 (and CD-1) mouse liver compared to CAST mouse liver (Figs 1A and S4E).

## Strain-shared and strain-unique sex-biased *cis* regulatory elements

We sought to identify regulatory elements for sex-biased liver gene expression, both those that are common and those that differ between B6 and CAST mice. We sequenced genome-wide ChIP-seq libraries for H3K27ac, which marks active enhancers and promoters, and we performed DNase-seq to identify open chromatin regions in liver nuclei collected from B6 and CAST mice of both sexes (Sheets B-D in S4 Table). These analyses identified thousands of regulatory elements (H3K27ac peaks and DNase hypersensitive sites (DHS)) in both strains and sexes (S5 Table). Using ChIP-qPCR and DNase-qPCR primers anchored at known sex-biased enhancers nearby strain-dependent sex-biased genes, we verified the presence of both B6-specific and CAST-specific epigenetic marks (S5 Fig). Furthermore, by using diffReps [73] to compare data for male vs female livers within each strain, followed by filtering to increase robustness (see Methods), we discovered ~2,000 robust sex-biased H3K27ac peak regions, and ~1,000 robust sex-biased DHS in livers of each strain (S6A–S6D Fig), with consistent sex bias seen across biological replicates (S6E Fig). A subset of these sex-biased regulatory regions was shared between strains and showed a concordant sex-bias, while many others were unique to one strain (Fig 2A and 2B). Thus, of 2,288 robust sex-biased B6 liver H3K27ac peak regions, 441 were shared with CAST mice, of which 98% showed concordant sex-bias (431 of 441; 198 female-biased, 233 male-biased) (Sheet E in S5 Table). Similarly, 92 sex-biased open chromatin regions (DHS) were shared between B6 and CAST mice, with 97% showing concordant sex-bias (89 of 92; Sheet D in S5 Table). Thus, many sex-biased regulatory regions maintain consistent sex-bias, despite the evolutionary divergence between B6 and CAST mice.

We use the term *cis* regulatory elements (CREs) to describe both strain-conserved and strain-unique regions with the potential to regulate genes in *cis*, i.e., intra-chromosomally or within the same Topologically Associating Domain (TAD) [74]. Overall, ~80% of the regulatory elements we identified—both sex-biased and sex-independent sequences—are outside of promoter regions (>2.5 kb), i.e., are enhancer regions (S6F Fig). Globally, strain-shared H3K27ac peaks and DHS regions showed greater sequence conservation across vertebrates than strain-unique peaks and regions (S6G Fig); however, the strain-shared sex-biased H3K27ac peaks and DHS regions did not show greater sequence conservation than the corresponding strain-unique regions. Thus, global sequence conservation alone does not explain the gain or loss of sex bias between strains.

Two examples of strain-shared sex-biased CREs neighboring core sex-biased genes are shown in Fig 2C: the male-biased complement component gene *C6* (16.4-fold average male

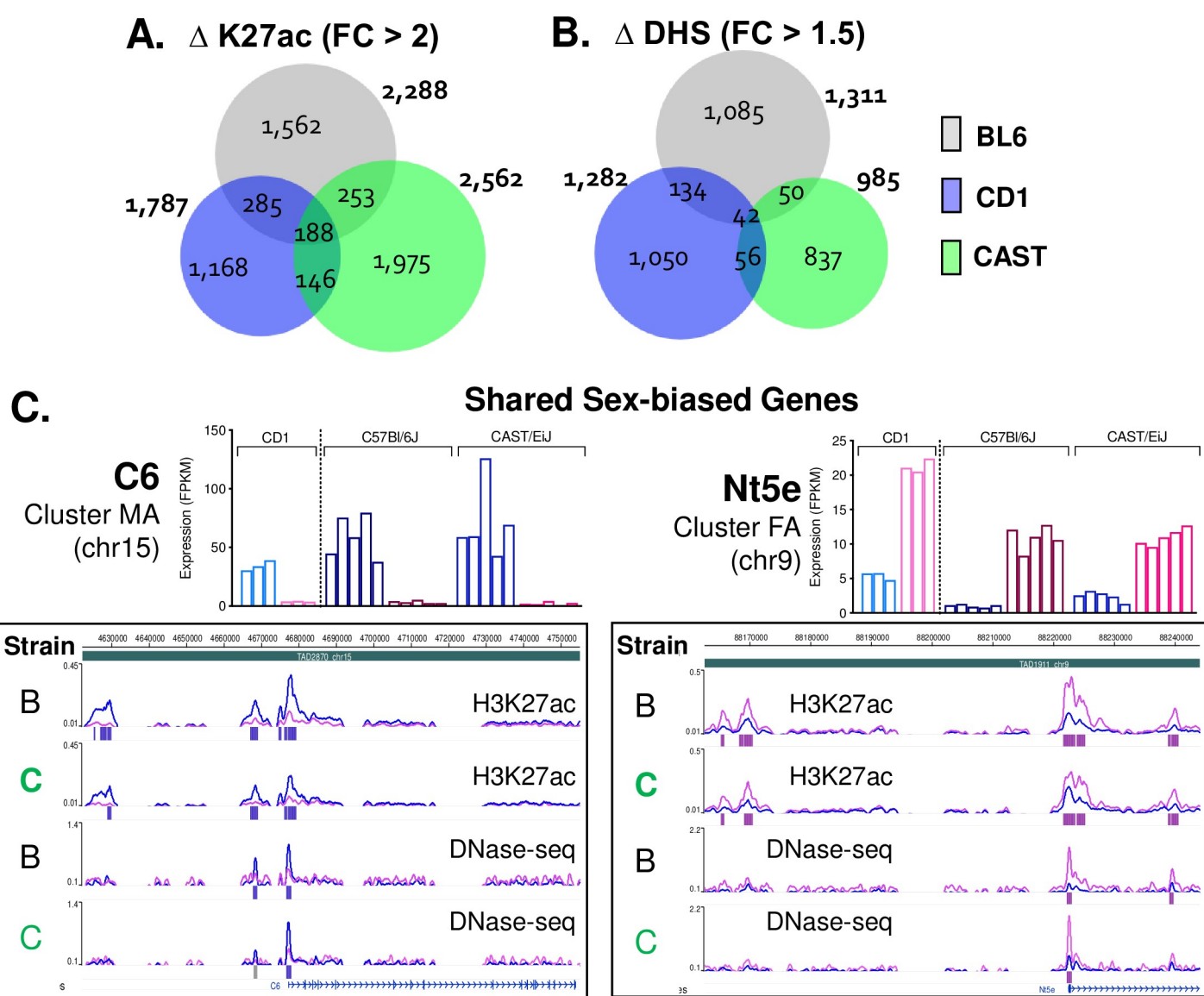

**Fig 2. Shared sex-biased enhancers. A, B.** Venn diagrams showing the number of sex-biased H3K27ac peaks (ΔK27ac) (>2-fold sex difference at FDR < 0.05, based on diffReps) or sex-biased DHS (ΔDHS) (>1.5-fold sex difference at FDR < 0.05, by diffReps) based on peak overlap by at least 1 bp. Data for CD-1 ΔK27ac peaks was provided by Dr. Dana Lau-Corona of this lab. CD-1 ΔDHS peaks were the robust sex-differential DHS dataset from [119]. **C.** Genes from clusters MA and FA (Fig 1A) with matching sex-biased *cis* regulatory elements in both strains. *C6* is a male-biased gene with male-biased H3K27ac marks and male-biased DHS at the proximal promoter and upstream enhancers. *Nt5e* is a female-biased gene with female-biased H3K27ac marks and female-biased DHS at both the proximal promoter and two neighboring enhancers. Bar graphs above the screenshots: RNA expression levels (FPKM) in individual male (blue bars) and female (pink bars) CD-1, B6, and CAST livers. Uniform expression across strains is observed, as expected for genes in clusters MA and FA. See Sheet A in S1 Table for full list of expression values.

bias across B6, CAST and CD-1 livers) and the female-biased nucleotidase *Nt5e* (5.7-fold average female bias). Extensive sex-bias, both locally at the promoter and at more distal enhancers, is seen for H3K27ac mark accumulation and chromatin opening with a directionality that matches the sex bias of gene expression. Other examples of core, strain-shared male-biased genes (*Gstp1*, *Elovl3*) and female-biased genes (*Acot3*, *Acot4*, *Cyp4a14*) showed uniform directionality in sex-biased enhancer activity and gene expression across strains (S7 and S8 Figs).

In closely related mammals, gene expression shows greater conservation than TF-binding or chromatin accessibility [44]. Consistent with this, a smaller fraction of sex-biased CREs was

conserved across strains as compared to the fraction of genes whose sex-biased expression is conserved across strains. Thus, while 26–33% of sex-biased genes are shared between B6 and CAST liver (Fig 1A), only 17–19% of sex-biased H3K27ac peak regions and 7–9% of sex-biased DHS are strain-shared (Fig 2A and 2B). This result can, in part, be explained by the complexity of enhancer coordination, where a conserved pattern of transcriptional output may emerge from the integration of the regulatory activity of multiple enhancers showing variability with regards to sex-bias of their activity. Supporting this idea, we observed both strain-shared and strain-unique sex-bias at several CREs nearby strain-shared sex-biased genes, such as the male-biased gene *Nudt7* and the female-biased gene *Cux2* (S7C and S8C Figs), which display a similar magnitude of sex bias between strains.

## Sex-biased CREs are closer to strain-shared than strain-unique sex-biased genes

Strain-shared sex-biased genes are generally closer than strain-unique sex-biased genes to CREs with matched sex bias. For example, 23.6% of all strain-shared sex-biased genes have a strain shared sex-biased CRE within 2 kb of their TSS, while only 15.2% of B6-unique sex-biased genes have a B6-unique sex-biased CRE within 2 kb of their TSS (Fig 3A vs. Fig 3B). The frequency of a strain-shared sex-biased promoter CRE is even higher (39%) for the 43 strain-shared sex-biased genes that show a consistent expression level in both strains (Fig 3A; genes in clusters FA or MA of Fig 1B). Both the strain-shared (Fig 3A) and the strain-unique sex-biased gene sets are more distant from sex-biased CREs of the opposite sex (Fig 3B and 3C; KS p < 0.0001). The strain-unique sex-biased genes were also significantly closer to strain-specific CREs in the same strain than similarly sex-biased CREs in the other strain (Fig 3B and 3C). After grouping sex-biased genes based on distance to correspondingly sex-biased CREs (proximal CRE, < 20 kb; distal CRE, > 20 kb but within the same TAD), we found that core, strain-shared sex-biased genes are more likely than strain-unique genes to be associated with a correspondingly sex-biased promoter region CRE (Fig 3D).

In a prior study, genes with strong liver sex bias were often subject to proximal regulation [52]. Consistent with this, strain-conserved sex-biased genes, with their greater frequency of proximal sex-biased CREs, showed a greater magnitude of sex-bias than strain-unique sex-biased genes (Fig 3E). However, the 43 strain-shared sex-biased genes with a strain-conserved expression pattern were not significantly more sex-biased than other 101 core sex-biased genes (M-W p = 0.698). Thus, the greater strength of sex bias for the strain-conserved genes cannot entirely be explained by their closer proximity to CREs.

## Enrichment of strain-specific genetic variants at sex-biased CREs

To investigate the impact of genetic variants on sex-biased gene expression, we compared the above sets of strain-specific CREs to a set of 1,172 eQTL regions implicated in the variation of sex-biased gene expression across a set of 438 male and female DO mouse livers [49]. Each eQTL region can be linked, with directionality of the eQTL's effects on sex-biased gene expression, to specific genetic variants originating from one or more of the eight DO mouse inbred founder strains. Two of the eight strains, CAST and B6, are the regulating strain (see Methods for definition) for 491 of the 1,172 eQTLs, which impact the expression of 481 sex-biased genes (Sheets A and B in S6 Table). Many more eQTLs and genetic variants are specific to CAST mice (406 eQTLs) than B6 mice (85 eQTLs) (S9A Fig), which is the only *m. castaneus* subspecies among the 8 DO mouse founder strains. The vast majority of eQTLs are *cis* to the affected gene: 78% (n = 383 eQTLs) mapped to the same TAD as the regulated gene and 14% (n = 67 eQTLs) were outside of the TAD but mapped to the same chromosome vs. only 8%

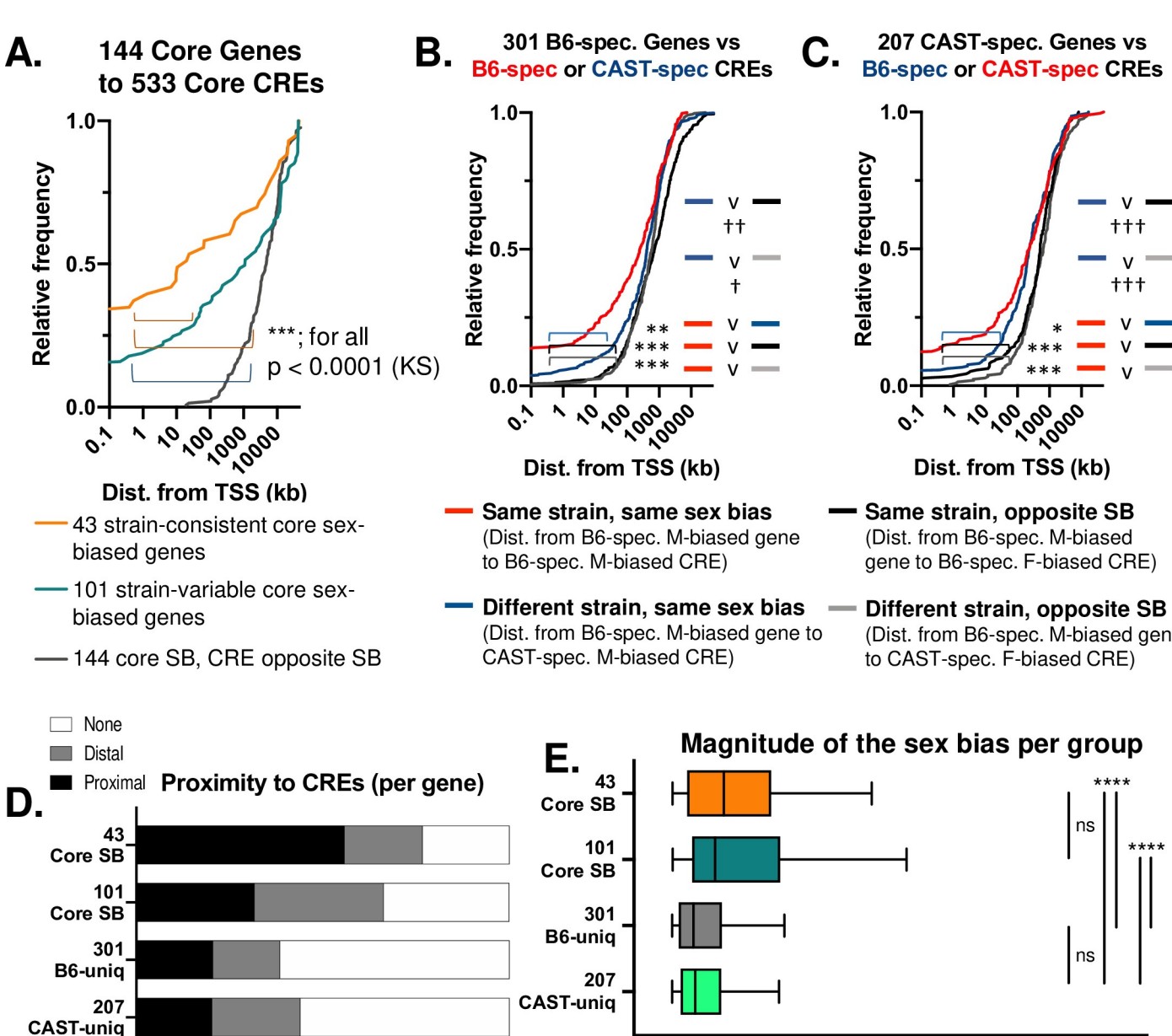

**Fig 3. Strain-shared and strain-unique sex-biased *cis* regulatory elements. A.** Cumulative frequency plots of distance from TSS of each gene set to the nearest core, strain-conserved sex-biased CRE with the same sex bias (i.e., distance from TSS of male-biased gene to the nearest male-biased DHS or male-biased H3K27ac peak). Plots are shown for 3 gene sets: 43 core sex-biased genes with strain-conserved expression (Fig 1B clusters MA or FA; orange), 101 core sex-biased genes with strain-variable expression (core sex-biased genes not in Fig 1B clusters MA or FA; teal), and the distance from all 144 core, strain-conserved sex-biased genes to the nearest sex-opposite CRE (e.g. distance from male-biased gene TSS to the nearest female-biased DHS or female-biased H3K27ac peak; gray). See also Sheet A in S1 Table. Significance was assessed by KS test: *, p < 0.05; **, p < 0.01; ***, p < 0.001; ns (not significant), p > 0.05. **B.** Cumulative frequency plot of distance from TSS of the 301 B6-unique sex-biased genes to four lists of *cis* regulatory elements (sex-biased DHS and sex-biased H3K27ac peaks): (1) B6-specific sex-biased CREs with the same sex-bias (as in panel A), in red; (2) CAST-specific sex-biased CREs with the same sex-bias, in blue; (3) B6-specific sex-biased CREs with opposite sex-bias, in black; (4) CAST-specific sex-biased CREs with opposite sex-bias, in gray. See Sheet C in S1 Table for B6-unique sex-biased genes. Significance, as in A. v, versus. **C.** Cumulative frequency plots of distance from TSS of 207 CAST-unique sex-biased genes to four lists of CREs (sex-biased DHS or sex-biased H3K27ac peak): (1) CAST-specific sex-biased CREs of the same sex-bias (as in A), in red; (2) B6-specific sex-biased CREs of the same sex-bias, in blue; (3) CAST-specific sex-biased CREs of the opposite sex-bias, in black; (4) B6-specific sex-biased CREs of the opposite sex-bias, in gray. See Sheet D in S1 Table. Significance, as in A. **D.** Core, strain conserved sex-biased genes are more proximal to similarly sex-biased CREs than strain-unique sex-biased genes. Shown are stacked bar charts representing percent of each group with a sex-biased enhancer of the same directionality that is TSS-proximal (< 20 kb; black), TSS-distal (> 20 kb but within the same TAD; gray) or lacking any similarly-sex-biased CREs within the same TAD (none; white). **E.** Box plot showing sex-bias each gene set group, shown as the absolute value of the log$_2$-transformed M/F fold-change values (EdgeR), based on data in S1 Table. Gene sets as in A-C. Significance, as in A.

(n = 41 eQTLs) mapped to a different chromosome (Fig 4A). In addition to relying on our prior characterization of eQTLs related to sex bias [49], we reanalyzed the underlying DO mouse datasets by adding a sex-eQTL interaction term into the model [75]. This analysis revealed that 168 of the 491 eQTLs with CAST or B6 as the regulating strain show a significant difference in LOD score between male and female liver (Sheet A in S6 Table). The genes regulated by these 168 eQTLs tend to exhibit greater sex bias than the genes regulated by the full set of 491 eQTLs (3.85 vs 2.25 median absolute fold-change; p = 6.12e-27), while the maximum expression was not significantly higher (p = 0.0831; S9B Fig). The subset comprised of 168 eQTLs was designated robust sex-specific eQTLs, while those only identified in the prior analysis, where each sex was treated separately [49], were designated lenient eQTLs.

One limitation of eQTL analysis is its comparatively low resolution (median interval = 1.7 Mb for 491 eQTLs for sex-biased genes) (S9C and S9F Fig). Consequently, the above set of 491 eQTLs impacting sex-biased genes encompassed a full 34% of all 10.2 million genome-wide SNPs/Indels between B6 and CAST mice (S9D Fig). Moreover, genetic variants differentiating CAST and B6 mice occur every ~100 bp [46] (every ~275 bp when considering variants unique to CAST or B6 mice relative to the six other DO founder strains; see Methods) and often occur within large, megabase-scale haplotype blocks, each containing thousands of SNPs and Indels. We therefore sought to significantly narrow the genomic region for consideration within each eQTL region. First, we limited our analysis to variants within the genomic region defined by the intersection of the eQTL region and the TAD containing the TSS of the regulated gene (intra-TAD interval, S9E Fig). This reduced the median interval size 4-fold, to 0.42 Mb, and decreased the number of genetic variants per eQTL region 3-fold, from 4,875 to 1,578 SNPs/Indels (S9F Fig). Second, we only considered as causative SNP/Indel candidates those that map to sex-biased CREs within the intra-TAD interval (c.f., [76]). This further reduced the median number of variants per eQTL by ~30-fold, from 1,578 to 51 (S9F Fig). We validated this approach by comparing the enrichment of strain-specific genetic variants at CREs within the intra-TAD interval of each eQTL to that of the full length eQTL (Fig 4B). Thus, we observed weak enrichment of strain-specific variants at sex-biased CREs when considering the full eQTL regions, but found a highly significant, 2.5-fold enrichment within intra-TAD intervals (p<0.001 for intra-TAD interval variants vs p = 0.278 genome-wide). In an important control, the enrichment in intra-TAD intervals was marginal when strain-specific variants at all CREs were considered. Furthermore, no enrichment of strain-specific variants at sex-biased CREs was found when we considered all CAST-specific or all B6-specific genetic variants genome-wide (1.01 observed/expected) (Fig 4B, first purple bar). Finally, variants were rarely found within a TF-binding motif (Fig 4B, green bars), consistent with prior reports [46,77,78].

## Strong genetic effects of DO mouse liver eQTLs reflect the gain or loss of *cis* regulation

286 of the above full set of 491 eQTLs regulating genes that exhibit sex-biased expression in B6 or CAST mouse liver can be categorized based on the mechanism whereby the eQTL either reduces or enhances sex differences in expression, as previously described [49] (S9G and S9H Fig). Specifically, eQTLs in categories #1–4 impart either a reduction in, or a total loss of, sex bias in the regulating mouse strain, while eQTLs in categories #5–8 confer either an increase in, or an outright gain of, sex-specificity in the regulating strain (Fig 4C) [49] (see Methods for definitions). These categories were further subdivided based on the sex in which the eQTL has a higher LOD score and the directionality of effect (e.g., Category #1 eQTLs result in repression of a male-biased gene in the regulating strain). Remarkably, 208 of the 286 categorized eQTLs (73%) resulted in either a loss, a gain, or a reversal of sex bias when comparing B6 and

**A.** eQTL (491) Position Relative to Regulated Gene (481 total)

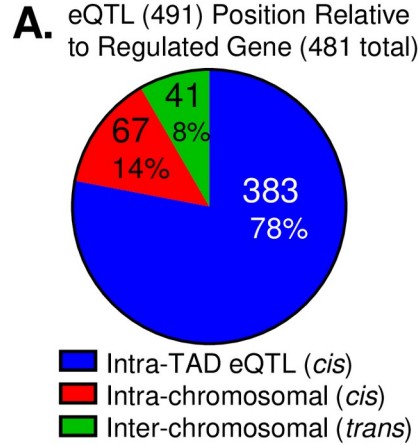

- ■ Intra-TAD eQTL (*cis*)
- ■ Intra-chromosomal (*cis*)
- ■ Inter-chromosomal (*trans*)

**B.** SNP/Indel Overlap with Features

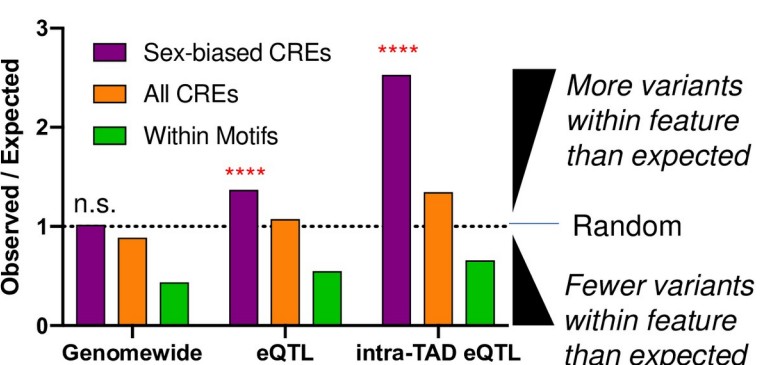

**C.** eQTL → **reduced** sex bias (cat. #1-4)

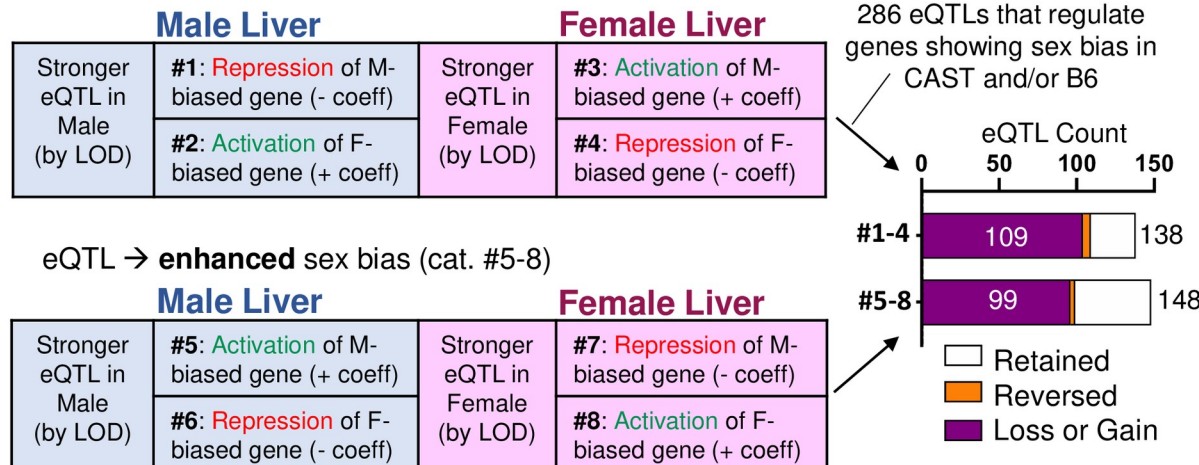

**D. Impact on gene's sex bias**

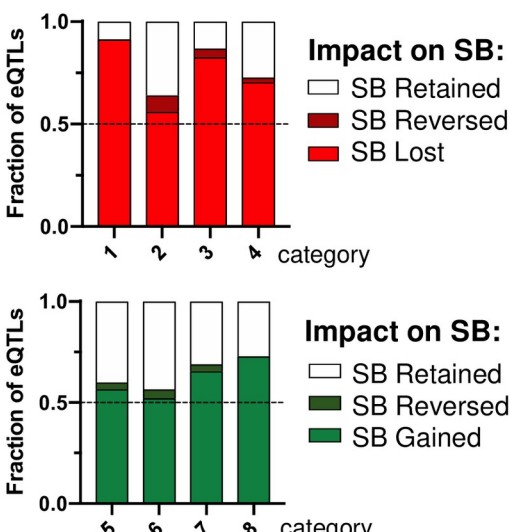

**E. Gain or loss of sex-biased CREs**

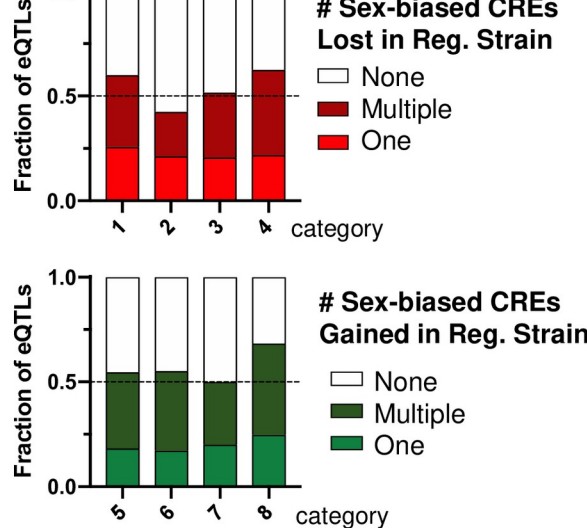

**Fig 4. Genetic determinants of strain differences in sex-biased gene expression integrating DO-eQTLs and CAST/B6 founders. A.** eQTL position relative to its regulated gene target for the 491 eQTLs where CAST or B6 is the regulating strain. eQTLs were placed in three groups: regulated gene in the same TAD as the eQTL (n = 383); regulated gene on the same chromosome but not overlapping in the same TAD as the eQTL (n = 67); and regulated gene on a different chromosome than the eQTL (n = 41). **B.** Genetic variants are enriched for sex-biased CREs within eQTLs. Shown is the enrichment (observed/expected) of the number of genetic variants within any sex-biased CRE (purple), all CREs (orange), or within any TF-binding motif. The number of genetic variants overlapping the indicated feature (observed) was divided by the expected number of overlaps for a length-matched, peak genomic position-shuffled random control (median of 1,000 permutations). A value of 1 indicates no enrichment. Example: the expected number of strain-specific genetic variants in intra-TAD eQTLs (383 of 491 eQTLs) that overlap sex-biased CREs is 1,767 (number of strain-specific SNPs/Indels in the regulating strain overlapping by at least 1 bp randomized genomic positions matched in size and length to the set of sex-biased CREs; median across 1,000 permutations). In contrast, the observed number of strain-specific genetic variants in intra-TAD eQTLs that overlap the actual positions of sex-biased CREs is 4,463. Thus, Observed/Expected = 4,463/1,767 = 2.53. **C.** Categorization of the 388 sex-dependent eQTLs significant in only male or only female DO mice and that are stronger in either male or female liver (based on difference in LOD score). eQTL categories #1–4 result in the loss or reduction of sex bias, while eQTL categories #5–8 result in enhancement or gain of sex bias (figure based on [49]). 388 of the 491 eQTLs can be categorized into groups #1–8. Of these 388 eQTLs, 286 regulate genes that are sex-biased in either B6 or CAST RNA-seq (S9G and S9H Fig; complete listing in Sheet A in S6 Table). *Right*: number of eQTLs that result in a complete loss or *de novo* gain of sex-bias (purple; sex bias is significant in only CAST or B6), reversal of sex bias (orange; sex bias is significant in both CAST and B6, but in opposite directions), or where sex-bias is retained, but at either an increased or a reduced level (white bars). **D.** Impact on sex-biased gene expression for the 286 eQTLs in categories #1–8 that regulate genes with significant sex bias (SB) in B6 or CAST mouse liver. Plots show the fraction of eQTLs in each category that result in a gain/loss, reversal, or retain sex bias but at a reduced level, as in panel C. **E.** Fraction of the 286 eQTLs that contain one, multiple, or no sex-biased CREs with genetic variants that plausibly explain the gain/loss of sex-biased gene expression (relevant CREs, defined in Methods). Relevant CREs for categories #1–4 are those lost in the regulating strain (with the same directionality as the regulated gene); relevant CREs for categories #5–8 are those that are gained in the regulating strain (with the same directionality as the regulated gene). See listing in Sheet A in S6 Table. The fraction of eQTLs within each category that can be explained by relevant sex-biased and strain-specific CREs ranged from 42% (category #2) to 68% (category #8). Results are nearly identical when considering all 388 eQTLs in categories #1–8, 223 (57%) of which contain relevant gain or loss of sex-bias at CREs (Sheet A in S6 Table).

CAST mouse liver (Fig 4C, right). Sex bias is retained but is either reduced (eQTL categories #1–4) or is increased (eQTL categories #5–8) in magnitude in the regulating strain for the other 78 eQTLs. The percent of eQTLs resulting in a gain or a loss of sex bias was as high as 91% for category #1 eQTLs (repression of male-biased genes in male liver) (Fig 4D). Mechanistically, 166 of the 286 eQTLs can potentially be explained by a strain-specific sex-bias in one or more CREs that harbor strain-specific genetic variants (Fig 4E and Sheet G in S6 Table). For eQTL categories #1–4, this is indicated by the absence in the regulating strain of one or more sex-biased CREs, which, in the other, non-regulating strain, show the same sex bias as the sex-biased gene. For eQTL categories #5–8, this is indicated by the presence in the regulating strain, but not in the non-regulating strain, of one or more sex-biased CREs showing the same sex bias as the regulated gene. Together, these findings, although correlative in nature, lend strong support for the proposal that the genetic factors identified by these 166 eQTLs impart the observed gain or loss of sex-biased gene expression primarily through the gain or loss of one or more of the associated sex-biased CREs. Specific examples, all robust sex-biased eQTLs, are presented for each eQTL category in Figs 5–8, below.

## eQTL-gene relationships where a single, potentially causal, sex-biased CRE is gained or lost

For 61 of the above 166 eQTLs, we identified only one sex-biased CRE that contains genetic variant(s) in the regulating strain, and which could thus be responsible for the observed gain or loss of sex bias (Sheets A and G in S6 Table). Examples include the male-biased genes *Cspg5* and *Olfm2* (Figs 5A and S10A) and the female-biased genes *Ptger3* and *Enpp1* (Figs 5B and S10B). *Cspg5* expression is 20-fold lower in CAST male liver, as compared to B6 male liver (category #1 eQTL), and only a single male-biased CRE, within the 5' UTR of *Cspg5*, contains CAST-specific genetic variants. Although other variants that are not CAST-specific are present in the region, all but two are outside of the indicated promoter-proximal CRE (Fig 5A; Unique versus All variants). The SNP within this CRE (G to C in CAST mice) is associated with the

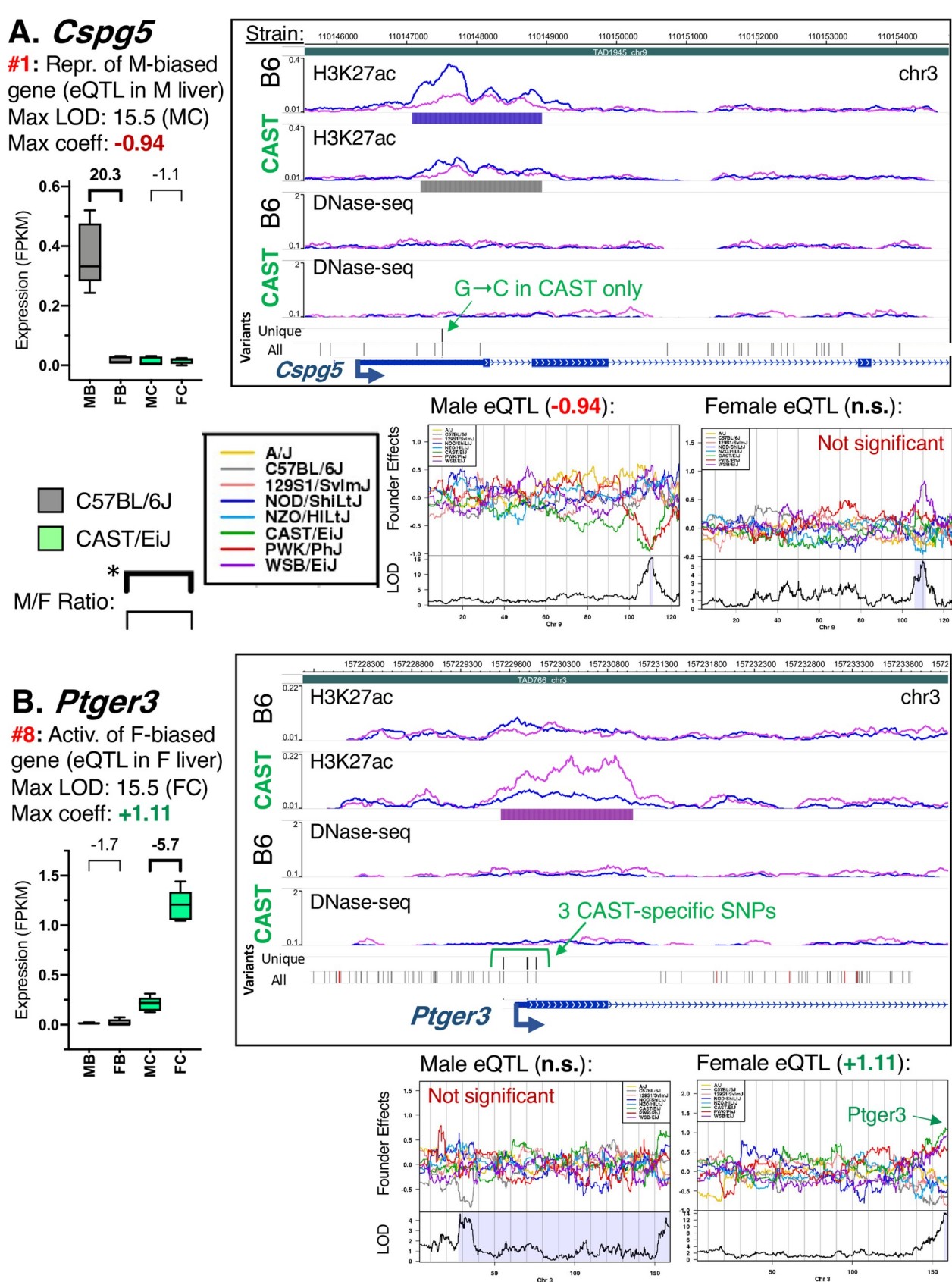

**A. *Cspg5***

**#1:** Repr. of M-biased gene (eQTL in M liver)
Max LOD: 15.5 (MC)
Max coeff: **-0.94**

**B. *Ptger3***

**#8:** Activ. of F-biased gene (eQTL in F liver)
Max LOD: 15.5 (FC)
Max coeff: **+1.11**

**Fig 5. Single CREs within eQTL that result in loss of male bias or gain of female bias.** Data is shown for *Cspg5* (**A**) and *Ptger3* (**B**). The format used here is also used in Figs 6–8. Genes and eQTL categories (Fig 4C) are listed along with the strain, sex, and regression coefficient for the regulating strain. Box plots present expression data (in FPKM values, from polyA+ liver RNA-seq; Sheets G and H in S1 Table) across four mouse liver groups: male B6 (MB), female B6 (FB), male CAST (MC), and female CAST (FC) (n = 5 livers per group). Above each box is the linear fold-change, where positive values indicate male/female expression ratios and negative values indicate female/male expression ratios. Significant sex bias (FDR < 0.05) is indicated by bold. WashU Epigenome browser screenshot (*right*) includes these tracks (top to bottom): H3K27ac ChIP-seq signal with peak regions identified for B6 and CAST liver; DNase-seq signal and DHS, with peak regions identified for B6 and CAST liver; genetic variants that fall within CREs from the regulating strain (either CAST or B6, as indicated in summary; Unique: strain-specific variants; All, all variants between B6 and CAST liver), and RefSeq genes. For ChIP-seq and DNase-seq signal tracks, normalized male sequence reads are indicated by blue tracings, female reads by pink tracings (tracings superimposed within each strain). Peak and DHS tracks (horizontal bars below signal tracks) indicate male-bias (blue bars), female-bias (pink bars), and any robust peaks/DHS without significant sex bias (gray bars). See S5 Table for a complete listing of peaks and DHS. Below each screenshot: regression coefficient plots across the chromosome harboring the eQTL, shown for all 8 DO mouse founder strains, with the most relevant data shown in green (CAST) and gray (B6), as indicated in the box. Positive regression coefficient: genetic variants from that strain in the indicated region are associated with higher gene expression of the regulated gene; negative regression coefficient indicates the opposite effect. Regression coefficient values for the regulating strain are indicated above the plot, calculated separately for DO male (*left*) and DO female liver (*right*). n.s. indicates the association is not significant for any founder strain in the sex indicated. If CAST or B6 is not the regulating strain in a given sex this is indicated in red above the regression coefficient plot (Not significant in CAST or B6). **A.** The male-biased gene *Cspg5* is repressed in CAST male liver, resulting in a loss of sex-biased expression in CAST liver (category #1 eQTL). This repression is associated with a single SNP (G to C in CAST) within the single male-biased peak in this eQTL that is lost in CAST mice (Sheet A in S6 Table). CAST is the regulating strain for this gene only in male DO mice (LOD = 15.5 in DO males; regression coefficient, -0.94). Significant male-biased expression is seen in B6 (20.3-fold M/F) but not in CAST liver. **B.** The female-biased gene *Ptger3* is activated in CAST female liver, resulting in a gain of sex-biased expression in CAST liver (category #8 eQTL). This activation is associated with three SNPs, all within the single female-biased peak in this eQTL that is lost in B6 mice (Sheet A in S6 Table). CAST is the regulating strain for this gene only in female DO mice (LOD = 15.5 in DO males; regression coefficient, +1.11). Significant female-biased expression is seen in CAST (5.7-fold F/M) but not in B6 liver.

loss of male-biased H3K27ac marks at the *Cspg5* promoter in CAST liver and loss of the 20-fold male bias in expression seen in B6 liver (Fig 5A). Conversely, the strong up regulation of the female-biased gene *Ptger3* in CAST female compared to B6 female liver (category #8 eQTL) is associated with a single female-biased CRE at the TSS and 5' UTR of *Ptger3* (Fig 5B). The three CAST-specific SNPs in this CRE are associated with a gain of female-specific H3K27ac marks and with up regulation of *Ptger3* expression in CAST female liver (5.7-fold female-bias in CAST liver vs no sex bias in B6 liver; Fig 5B). For both *Cspg5* and *Ptger3*, the sex-biased CREs shown are the only candidate regulators harboring strain-specific SNPs within the eQTL region (Sheet A in S6 Table).

Enhancement of the sex-bias of expression can also be associated with loss of positive regulatory elements (eQTL categories #6 and #7). Thus, the female-biased expression of *Slc16a5* increases from 5.5-fold in B6 mouse liver to 48.7-fold in CAST liver, primarily due to a decrease in basal expression in CAST males (category #6 eQTL; Fig 6A). *Slc16a5* expression, DHS, and H3K27ac marks at the *Slc16a5* promoter are equivalent in B6 and CAST female liver, however, all three are reduced in CAST male liver compared to B6 male liver in association with CAST sequence variants (Fig 6B). Conversely, the repression of the male-biased gene *Bok* seen in female CAST liver (coefficient -0.91) but not in male CAST liver results in an increase in male bias (S10C Fig). This repression in female CAST liver can be explained by the decrease in H3K27ac marks within the *Bok* gene body in female CAST liver compared to female B6 liver (S10D Fig), and consequently, there is a female-biased CRE in CAST but not B6 liver (category #7 eQTL; S10C Fig). These examples illustrate how genetic variants within a single CRE may lead to a gain or loss of sex-biased expression.

## Coordinated gain or loss of sex bias across multiple CREs within an eQTL

A majority of the eQTLs in each category show a gain or loss of multiple sex-biased CREs (Fig 4E; 105 of the 166 sex-biased eQTLs discussed above), and hence their regulation and the genetic mechanisms governing the strain-dependence of their sex-biased expression is intrinsically more complex than the cases of single sex-biased CREs discussed above. Indeed, as illustrated by the examples below, CREs within these eQTLs typically showed coordinated, strain-

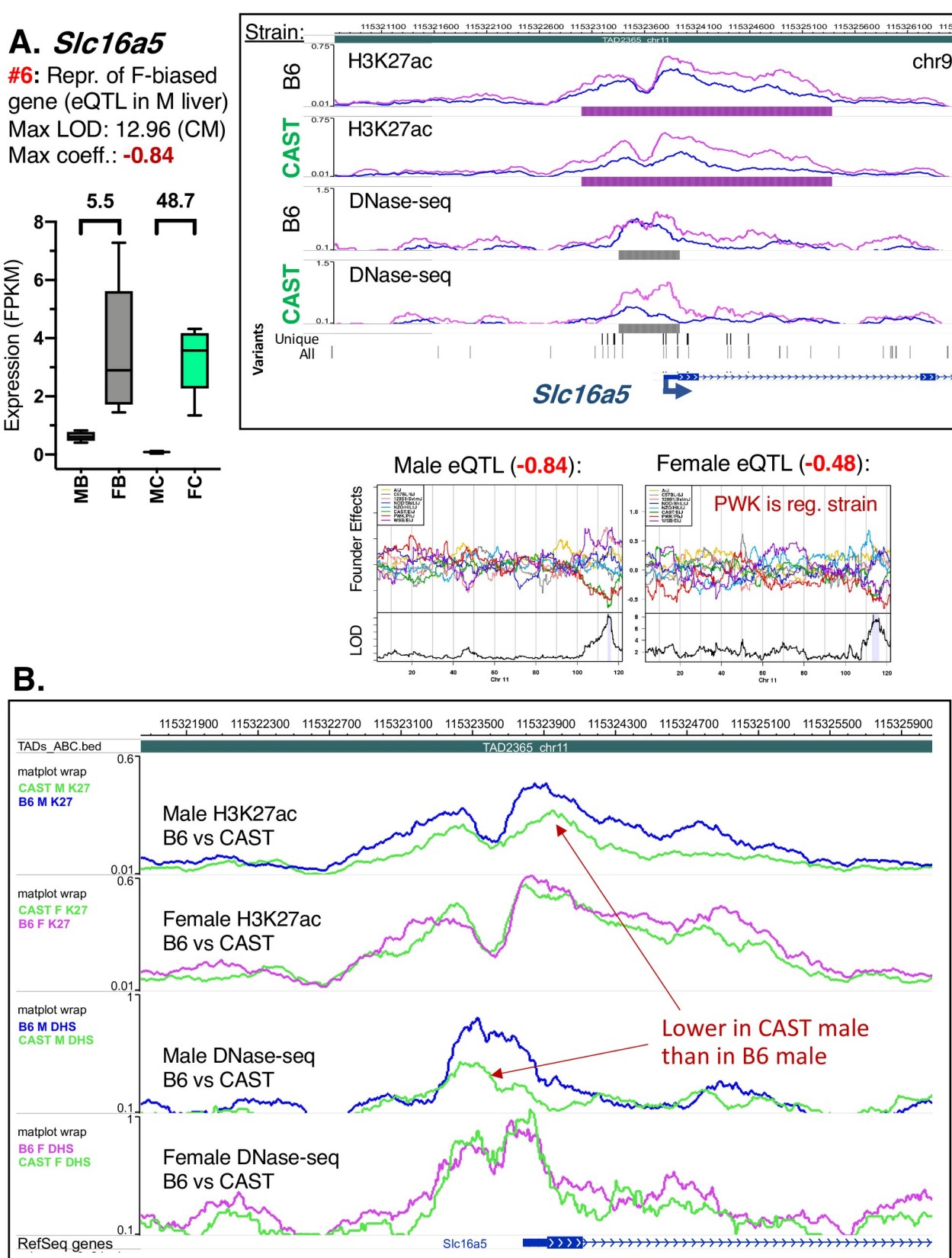

**A.** *Slc16a5*

**#6**: Repr. of F-biased gene (eQTL in M liver)
Max LOD: 12.96 (CM)
Max coeff.: **-0.84**

**Fig 6. Enhanced sex-biased expression due to gene repression in the opposite sex. A.** The female-biased gene *Slc16a5* is repressed in CAST male liver, resulting in a robust enhancement of female-biased expression in CAST liver (category #6 eQTL). This repression is associated with multiple CAST-specific genetic variants within a strain-shared female-biased peak in the promoter of *Slc16a5* that extends into the first intron. CAST is the regulating strain for this gene only in male DO mice (LOD = 12.96 in DO males; regression coefficient, -0.84), while PWK is the regulating strain in female DO mice. The significant female-biased expression seen in CAST liver (48.7-fold F/M) is retained, but at a reduced level in B6 liver (5.5-fold F/M). Browser screenshot shows normalized sequence read signal tracks superimposed by strain. Data presented as in Fig 5. **B.** Browser screenshot presenting normalized sequence read signal tracks superimposed by sex. This allows direct comparison of normalized sequence read abundance between B6 and CAST mouse liver of the same sex. A reduction of H3K27ac mark accumulation in CAST male liver at the promoter of *Slc16a5* enhances the female bias of the CRE (as is characteristic of category #6 eQTL). B6 tracings are shown in blue (male) or pink (female); CAST tracings are shown in green for all tracks.

dependent differences in their sex-biased enhancer activity, which in many cases cannot be explained by a strain-specific SNP or Indel within the CRE.

*Moxd1* shows a 12-fold increase in male bias in CAST compared to B6 liver (from 64-fold to 776-fold), due to a large increase in expression in CAST male liver (Fig 7A, category #5 eQTL). This increase is associated with increased activity of multiple male-biased CREs—both H3K27me3 sites and DHS—in CAST but not B6 male liver (Fig 7B). These strain-specific, sex-dependent CREs are found at the *Moxd1* promoter and at enhancer regions e1 and e2. Importantly, many but not all the CAST-specific, male-biased CREs at enhancer e2 contain strain-specific SNPs or Indels (Fig 7B; red asterisk, regions lacking such SNPs/Indels). Two downstream enhancer regions, e3 and e4, contain male-biased CREs in both CAST and B6 liver, and may account for the basal level of male-biased *Moxd1* expression seen in B6 liver. Positive regression coefficients for the CAST genetic background were seen in both male and female DO mice; however, the effect is stronger in males (+1.68 in males vs. +0.89 in females; Fig 7A). In a corresponding example of a female-biased gene, *Bmper*, we observed strong activation in CAST female liver (category #8 eQTL) associated with multiple nearby female-biased CREs (S11 Fig). Two female-biased intronic enhancers appear to be sufficient to maintain a 5-fold female bias in the expression of *Bmper* seen in B6 mouse liver; however, the activation of 9 additional female-biased CREs in CAST liver increased the magnitude of female bias to 17.4-fold in CAST liver (S11 Fig).

S12 Fig presents a case where multiple CREs at or near a cluster of three male-biased genes (*Rassf3*, *lnc9349*, lnc9351) with overlapping eQTLs coordinately lose male bias due to the activation of their expression in CAST female liver (category #3 eQTL). This genomic region contains multiple male-biased CREs in B6 liver, all but one of which show increased enhancer activity (H3K27ac marks and/or DHS) in CAST female liver, with no change in CAST male liver. Thus, there is a coordinated loss of nearly all male-biased CREs over a wide (~35 kb) genomic region in CAST liver, driving the increased expression seen in the CAST females (S12B Fig). This effect is strongly sex-dependent and strain-dependent, with strong positive regression coefficients seen in CAST female but not CAST male liver for all 3 genes (S12C–S12E Fig). In this case, however, we also need to consider non-strain-specific SNPs/Indels between CAST and B6 mice in the eQTL region, insofar as genetic effects were also observed in a second strain, PWK, and where the promoter region of *lnc9349* includes a sex and strain-dependent CRE that harbors two SNPs that are shared between CAST and PWK mice but absent in B6 mice (S12B Fig, red arrows).

Finally, we present a ~125 kb genomic region within a single TAD on chr 10 with overlapping eQTLs for three female-biased genes, which all lose their sex-biased expression in either B6 liver (*Gm4794* (*Sult3a2*)) or CAST liver (*Sult3a1* and *Rsph4a*), albeit by different mechanisms (Fig 8). *Gm4794* (*Sult3a2*) and *Sult3a1* are both regulated by category #2 eQTLs (loss/decrease in female bias due to gene activation in male liver; c.f., higher regression coefficients in DO male liver than DO female liver, whereas *Rsph4a* is regulated by a category #4 eQTL

**A.** *Moxd1* Activation of M-biased gene in **CAST** male liver (cat. **#5**)

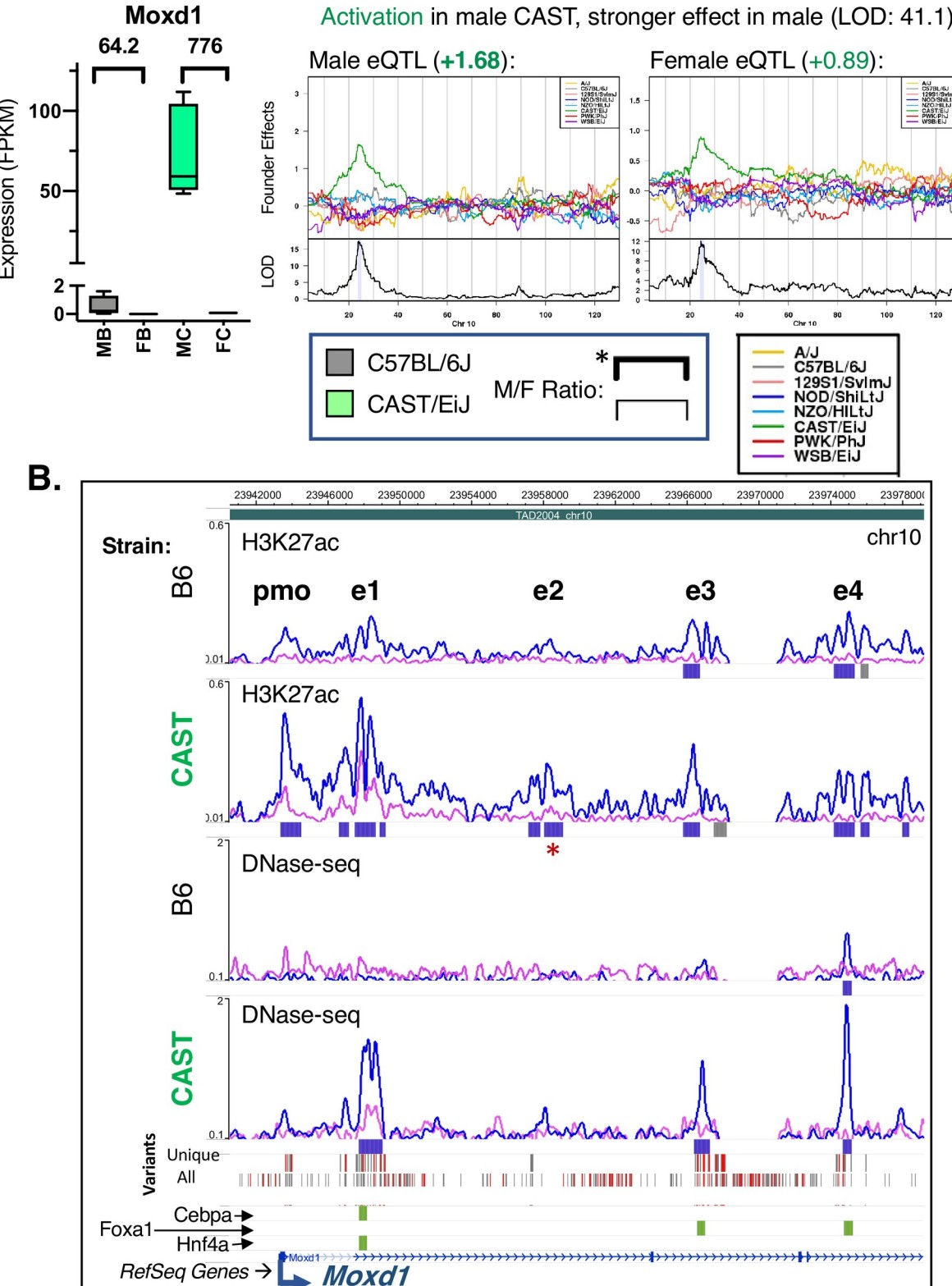

**Fig 7. Coordinated activation of multiple CREs leads to increased male bias. A.** The male-biased gene *Moxd1* is regulated by a stronger, positive eQTL in male compared to female CAST liver, increasing its male-biased expression in CAST compared to B6 liver (category #5 eQTL). This activation is associated with multiple CAST-specific genetic variants found within several CREs, including both strain-shared male-biased peaks and CAST-unique male-biased peaks. CAST is the regulating strain in both female and male DO liver, but with a stronger effect in male. The significant male-biased expression in B6 (64.2-fold F/M) is substantially enhanced in CAST liver (776-fold F/M; sex bias retained, and at a higher level). Y-axis is split to highlight the sex-biased expression seen in B6 liver. Data presented as in Fig 5. **B.** Browser screenshot showing 5 CREs at the promoter (pmo) and at four enhancers (e1 through e4) at *Moxd1*. Given many genetic variants in this region, gray bars indicate single variants and red indicates multiple variants (2 or more) in the CAST Variants track. Three other tracks indicate the strain-specificity of transcription factor binding (green: CAST-preferential; orange: B6- preferential) for Cebpa, Foxa1, and Hnf4a (top to bottom). Red asterisk: H3K27ac peak with the relevant pattern of sex bias (gain of male bias in CAST liver) that lacks a strain-specific SNP/Indel (see Discussion). Data presented as in Fig 5.

(loss of female bias by repression in female liver) (S13A–S13C Fig). *Gm4794* (*Sult3a2*) and *Sult3a1* both show striking increases in CREs (both H3K27ac marks and DHS) and in gene expression in CAST livers from both sexes, which introduces a sex bias in expression in the case of *Gm4794* (*Sult3a2*) and abolishing one in the case of *Sult3a1* (Fig 8A). The loss of female-biased expression of *Sult3a1* occurs, even though many of the strain-specific CREs within its eQTL region acquire female-biased H3K27ac marks and/or female-biased DHS in CAST liver (Fig 8B). Strikingly, many of the strain-dependent, sex-specific CREs seen in CAST liver do not overlap strain-specific variants (red asterisks, Figs 8B and S13D).

The eQTL regulating *Rsph4a* is only significant in female DO mouse liver (-1.01 regression coefficient, CAST-specific; S13C Fig), and consistent with that, *Rsph4a* is strongly repressed in female CAST liver compared to female B6 liver, abolishing its female-specific expression pattern (Fig 8A). The only B6-specific female-biased CRE in this gene region is at the promoter of *Rsph4a* and overlaps a CAST-specific SNP (S13E Fig). Sequence variants that disrupt TF motifs are rare, and represent only 1–5% of genetic variants [46]; however, this CAST-specific SNP at the female-biased *Rsph4a* promoter CRE, seen in B6 but not CAST liver, is one such example: it disrupts an HNF6/Cux2 motif within an HNF6 ChIP-seq binding site identified in CD-1 mouse liver [79] and it likely contributes to the observed loss of female-biased *Rsph4a* expression.

## Discussion

We elucidated both genetic and epigenetic factors that contribute to the sex-biased patterns of hepatic gene expression and *cis* regulatory activity in two evolutionarily distant mouse strains, B6 and CAST. We characterized both strain-shared and strain-unique sex-biased genes, including many lncRNA genes. Strain-shared sex-biased protein-coding genes were found to be highly liver-specific in their expression and biological functions and were regulated by TSS-proximal enhancers, whereas sex-biased genes unique to each strain were significantly less liver-specific and more often regulated by distal enhancers. We also identified hundreds of *cis* regulatory elements (CREs, primarily enhancers) with a strain-shared sex-bias, and we elucidated the strain-dependent gain or loss of many other strain-specific, sex-biased CREs. A subset of the strain-specific CREs harbored strain-specific SNPs/Indels, which we linked to eQTLs regulating the strain-dependent, sex-biased expression of individual sex-biased genes. In many cases, these eQTL-gene relationships reflected the gain or loss of a single, apparently causal, strain-specific sex-biased CRE. Remarkably, in many other cases we observed a coordinated gain or loss of sex bias across multiple CREs within an eQTL, including CREs that did not contain strain-specific SNPs or Indels, which raises fundamental questions about underlying mechanisms. Thus, by integrating biological data at the genetic, epigenetic, and transcriptomic levels across divergent inbred mouse strains, we harnessed the power of natural genetic diversity to gain novel insights into the biology and genetics of liver sex differences, including the

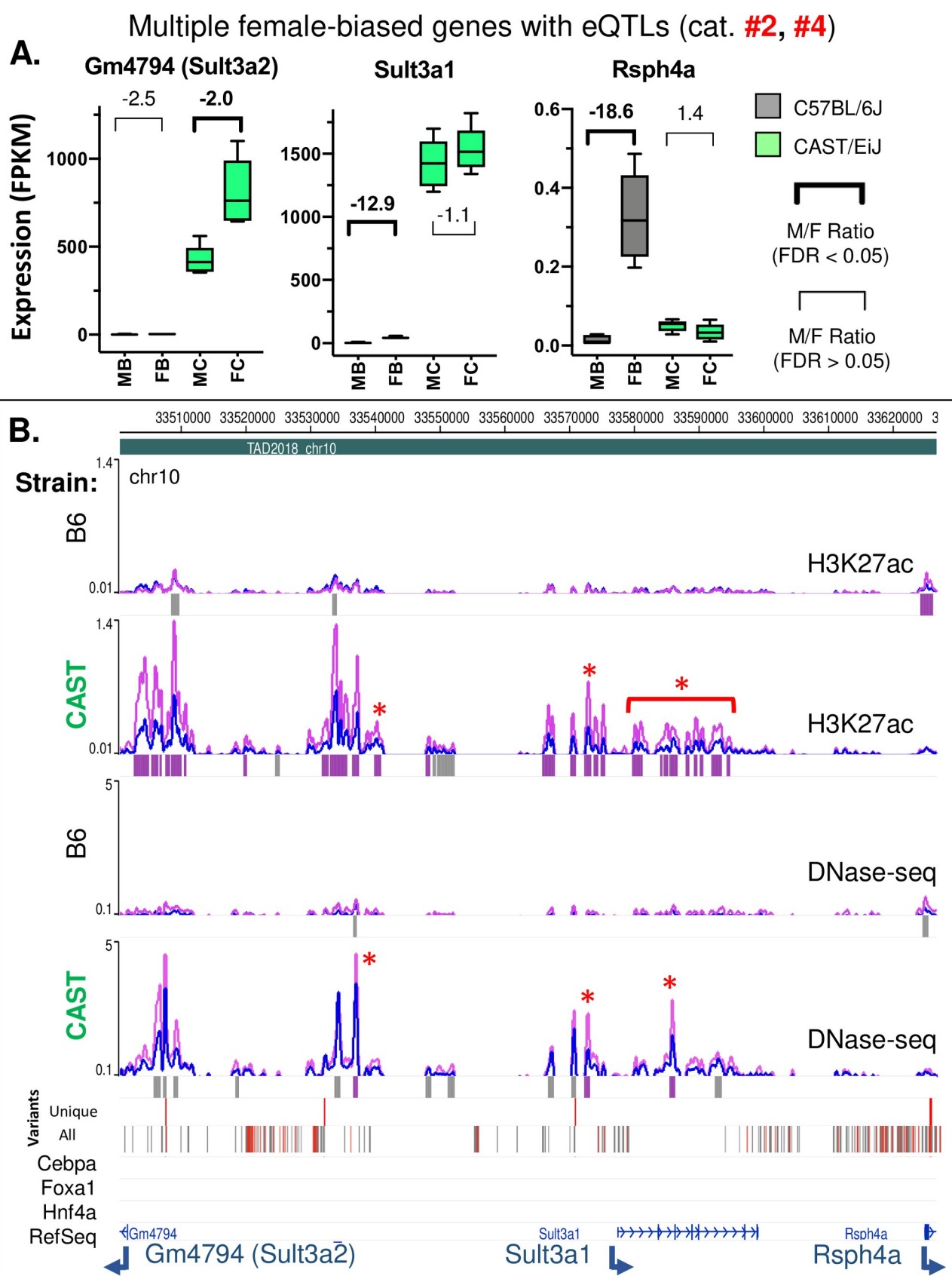

**Fig 8. Overlapping eQTLs with different modalities leads to reduction or loss of female bias by different mechanisms. A.** Expression of the female-biased genes *Gm4794/Sult3a2* and *Sult3a1* is strongly increased in CAST liver, with a stronger genetic effect in males (see regression coefficients, S13A and S13B Fig). This reduces the magnitude of the female bias in expression (category #2 eQTL). The male-biased gene *Rsph4a* is repressed in female CAST liver, resulting in a loss of female-biased expression (significant only in DO female, category #4 eQTL; see S13C Fig). Data presented as in Fig 5. **B.** Browser screenshot showing strong increase in CRE activity in CAST compared to B6 liver, with multiple CAST-specific female-biased CREs, and a single female-biased CRE unique to B6 at the promoter of *Rsph4a*. Horizontal bars beneath the CAST Variants track: grey, single strain-specific variants; red, multiple variants. Data presented as in Fig 5. Additional zoomed in screenshots for *Sult3a1* and *Rsph4a* regions are shown in S13D and S13E Fig.

discovery of hundreds of genomic regions implicated in sex-biased gene regulatory mechanisms.

We identified a core set of 144 protein-coding genes whose sex-biased expression is conserved between B6 and CAST mouse liver. Many are liver-specific genes with strong enrichment for molecular function gene ontologies related to liver physiology. The conservation of sex-bias for these genes across the substantial evolutionary distance between CAST and B6 mice suggests they play a fundamental role in the biology of sex differences in the liver and its regulation. One example is the conservation between strains of the sex-biased expression of Bcl6 and Cux2, two GH-regulated transcriptional repressors (TFs) [71,80] that play a critical role in enhancing the sex-bias of many genes in mouse liver [40,41,79]. We also identified 78 strain-shared sex-biased lncRNAs, which may play regulatory roles in the chromatin-bound fraction either in *cis* or in *trans* [18,81,82]. Four of these 78 lncRNAs have been implicated in negative regulation of sex-biased protein-coding genes of the opposite sex bias and inverse hypophysectomy response class (lnc630, lnc2937, lnc7423, lnc9000; Sheet B in S1 Table) [17]. In contrast, three other lncRNAs implicated in negative regulation of other sex-biased genes [17] showed sex-biased expression in B6 but not CAST liver, and could contribute to the strain differences in sex-biased gene expression described here.

Three TFs showed sex-biased expression in B6 but not CAST mouse liver: Tox, Trim24 (both female-biased) and Zfp809 (male-biased). All three TFs are subject to GH-regulated expression [13,71,72] and may contribute to the strain differences in sex-biased gene expression via *trans* regulation. Tox is a regulatory factor in T-cell exhaustion [83] and is a therapeutic target for cancer immunotherapy [84], but its role in gene expression in hepatocytes is unclear. Zfp809 is a KRAB-zinc finger protein implicated in epigenetic silencing of endogenous retroviral sequences [85]. The male bias in *Zfp809* expression seen in B6 but not CAST mouse liver could thus impart strain differences for genes where endogenous retroviral silencing is critical. Incomplete silencing of endogenous retroviral sequences contributes to strain differences in spontaneous and drug-induced liver tumorigenesis [86] but has not been explored in the context of sex differences. *Trim24* is a suppressor of mouse hepatocellular carcinoma [87,88]; consequently, its elevated expression in female compared to male B6 mouse liver may contribute to the lower incidence of liver cancer in females [70,89]. Our finding that the female bias in *Trim24* expression is lost in CAST mouse liver, where *Trim24* shows high expression in males (S2 Fig), suggests sex differences in hepatic tumorigenesis may be reduced in CAST mice. B6 and CAST mice differ in their sensitivity to genotoxic damage induced by 1,3-butadiene, which induces strain-dependent epigenetic and transcriptomic changes in liver, lung and kidney [90,91]; however, differences in hepatocellular carcinoma between mouse strains [92,93] have not been determined for B6 vs CAST mice.

We recently identified a large number of eQTLs that regulate the expression of sex-biased gene expression in mouse liver [49], but little is known about how these autosomal genetic factors impart their regulatory effects. Adding to the problem, the loci identified are large and often contain thousands of SNPs and Indels. Here, we show that many of these eQTLs can be linked to the strain-specific gain or loss of sex-biased chromatin opening and activating

enhancer marks at CREs within the eQTL regions, a subset of which harbor strain-specific SNPs/Indels. Thus, 208 of 286 eQTLs with apparent sex-differential effects on sex-biased gene expression showed a complete gain, loss, or reversal of sex-biased expression for the regulated gene. Further, 166 of the eQTLs were associated with a gain or loss of at least one sex-biased CRE harboring strain-specific genetic variants. Importantly, we found that 61 of the eQTL regions contain a single relevant CRE with genetic variants that impact sex-biased expression, which has a high likelihood of being causal for the sex-dependent effects of the eQTL on sex-biased gene expression. Further, for 105 other eQTLs, strain differences in expression were linked to a coordinated gain or loss of multiple strain-dependent, sex-biased CREs within the eQTL region. While we cannot conclusively distinguish a primary, causal enhancer CRE from secondary CREs within a given eQTL based on our study design, in many cases the relevant strain-specific CREs lacked strain-specific genetic variants, but nevertheless showed a gain or loss of sex-bias coordinated with other, neighboring CREs harboring strain-specific variants, indicating they are likely responding in a secondary manner to a causal, strain-specific genetic event.

It is a major challenge to ascertain the contributions of the individual, constituent CREs, and any causal association with gene expression, for eQTL regions that contain multiple CREs with strain-specific SNPs. For such eQTLs, genetic variants within one of the multiple eQTL CREs may be the primary causal driver of the observed gain or loss of sex-biased enhancer activity and gene expression across the eQTL. For example, a SNP/Indel that activates a single causal, strain-specific CRE may nucleate activity at neighboring CREs that lack an activating sequence variant, as occurs in T-cell leukemias, where a mutation at a single oncogenic enhancer activates multiple other enhancers and up regulates the *TAL1* oncogene [94]. Cooperativity in enhancer activation of proximal genes is well established [95], however, the exact nature of the relationship between clusters of enhancers and gene targets remains controversial [96]. In some cases, deletion of single enhancers leads to a dramatic loss of activity at neighboring enhancers, suggesting a hierarchical relationship [94,97,98], with a clear delineation between hub and non-hub [99] and between predominant and supportive enhancers [98] within larger enhancer clusters. However, other studies find that enhancers contribute mostly additively to gene expression, with redundancy between individual enhancers [100,101]. Indeed, there may be fundamental biological differences between enhancer clusters, only some of which exhibit hierarchical 3-dimensional structure [99]. Even within a single enhancer cluster, hierarchical relationships may vary, depending on hormonal cues and downstream signaling [97,102]. Further work, including careful dissection using genetic editing in mouse liver *in vivo*, will be needed to determine the respective contributions of individual sex-biased CREs whose strain-dependent gain or loss of activity is coordinated with neighboring regulatory elements, and to elucidate the underlying features that determine whether a given eQTL region best fits a hierarchical model or an additive model of enhancer coordination.

None of the strain-specific genetic variants presented here, except one at the eQTL for *Rsph4a*, directly disrupts a TF motif. Indeed, we found that strain-specific genetic variants are depleted from intra-motif coordinates relative to randomized genomic coordinates, consistent with prior reports [45,46,77,78]. Disruption of enhancer activity may occur by a combination of effects on DNA shape, nucleosome positioning, and co-binding TFs [103]. Liver sexual dimorphism requires the coordinated participation of multiple TFs, both those that are sex-biased and/or GH-responsive, such as STAT5 [40,41,57,79], and sex-independent TFs essential for many liver-expressed genes and functions [37,70]. This coordination is likely to involve TF-TF interactions within large, co-bound TF clusters, which are stabilized by multiple suboptimal motif-adjacent sequences [104]. While genetic variants outside of canonical binding motifs may have modest effects on the binding of individual TFs, they may nevertheless

destabilize the overall multi-TF clusters that form [105]. Consistent with this idea is the coordinated gain or loss of multiple strain-specific, sex-biased CREs that we found at 105 eQTLs regulating the strain-specific expression of sex-biased genes.

Our comparison of sex-biased gene expression and sex-biased regulatory elements across mouse strains has some limitations. We investigated the impact of strain-specific genetic variation on the activity of CREs, but did not consider genetic differences at transcribed sequences, which can impact RNA stability [106]. Furthermore, genetic variants may contribute to differences in 3D genome organization [107–109], which may differ between B6 and CAST mouse liver. Our mapping of sex-biased CREs to eQTL target genes has already taken into account genome structure in the form of TADs, which define broad regulatory domains that restrict enhancer-promoter interactions [110,111] and significantly enriched for genetic variants at sex-biased CREs as compared to full-length eQTL regions (Fig 4B). However, we did not consider potential strain differences in TAD or intra-TAD structure [76,112,113], which may allow *cis* regulation to be defined with greater precision [74]. Moreover, strain differences in DNA looping may be independent from the strain differences in epigenetic marks of enhancer activity that we measured, and consequently, some of the strain-shared sex-biased CREs identified here may interact with their sex-biased gene targets in a strain-specific manner. Chromosome conformation assays have revealed sex-dependent looping interactions in mouse liver [54] and may be useful for evaluating strain-specific sex bias at the level of DNA looping.

The nomenclature of "gain" and "loss" of sex bias used throughout this study is based on the underlying assumption that the private/unique genetic variants that are the focus of this study are unlikely to be shared between the regulating strain and the ancestral state (see Methods for additional definition of these terms). While we consider this assumption most probable, it remains a possibility that the ancestral allele and even an allele that is strain-specific among DO founder strains are shared. This is more likely for CAST mice than for B6 mice, given its evolutionary distance from the other founder strains. Further analysis of several additional wild-derived mouse strains more closely related to CAST would be required to further evaluate this issue.

Our work is also limited by its focus on only two mouse strains, B6 and CAST. In particular, by focusing this study on strain-specific genetic variants and by strictly requiring that these variants be unique across all 8 DO founder strains, our analysis explicitly excludes variants shared between B6, or CAST, and one or more other founder strains. Of note, this exclusion has the beneficial effect of greatly limiting the number of SNPs to be considered within an eQTL region (e.g., Fig 5, Unique vs All variants track). Moreover, this approach is appropriate for the 70–74% of eQTLs that do not involve major contributions from a second strain (125 of the 168 robust sex-biased eQTLs; and 344 of all 491 eQTLs; Sheet A in S6 Table), and as a result, we consider it less likely that non-strain specific SNPs account for the strain differences in expression between B6 and CAST mice for these eQTLs. Further, as noted above, we cannot validate our assignments of loss and gain of sex-specific gene expression alleles without knowing the true ancestral allele. A loss of gene expression pattern in one strain could be a gain of gene expression pattern in multiple DO founder strains, a case more likely to occur when the mutation is acquired prior to a speciation event, for example, speciation of the six *M. m. domesticus* strains examined here.

Finally, while a majority of intra-population variation in TF binding or chromatin state is attributable to *cis*-acting genetic variants [46,114], genetic factors may also impact sex-dependent gene expression through strain-specific *trans*-acting factors [115]. Indeed, we identified 41 *trans*-acting eQTLs that affect sex-biased gene expression, nine of which target sex-biased lncRNAs (Sheet A in S6 Table), which themselves may act in a *trans* manner. In addition, a subset of the strain-unique, sex-specific CREs that we identified is likely not the direct result of

local genetic differences, but rather may reflect a loss in CAST mouse liver of sex differences in the expression of their bound transcriptional regulators, such as Tox, Zfp809 or Trim24, as noted above. An F1 cross of CAST x B6 mice could address this question by placing a copy of each genome in the same *trans* environment, and thereby help differentiate *cis* from *trans* effects on gene expression and transcription factor binding [46,51]. Furthermore, it is difficult to ascertain lineage-specific gain or loss of sex-bias without an outgroup such as *M. Spretus* (~2MY diverged from CAST and B6 mice; [116]), for which gene expression and CRE data (TF binding activity) is only available for male liver [45]. Thus, the terms gain and loss of sex-bias, as used here, may not represent the full evolutionary history of murine subspecies.

In conclusion, we report strong genetic effects on the expression of sex-biased genes in a mouse liver model, with several hundred eQTLs resulting in a complete gain, loss, or reversal of sex-biased expression between B6 and CAST mice. We identified at least one CRE that gains or loses sex bias (either chromatin opening or H3K27ac mark accumulation) in a manner that can explain the gain or loss of sex-biased gene expression for a majority of these eQTLs. In many cases, we observed the coordinated gain or loss of sex bias across multiple CREs within an eQTL, which makes the elucidation of causality of individual genetic variants difficult. Despite the widespread occurrence of these strong genetic factors, we identified a core set of strain-shared sex-biased genes, including many lncRNA genes, with liver-specific patterns of expression and molecular functions. We can anticipate that our findings, and the approaches taken in this investigation of the role of genetics and epigenomics in liver sex differences, will be critical for translating studies from mouse models to humans.

## Methods

### Ethics statement

Mice were treated using protocols specifically reviewed for ethics and approved by the Boston University Institutional Animal Care and Use Committee (protocol approval # PROTO201800698), and in compliance with ARRIVE 2.0 Essential 10 guidelines [117], including study design, sample size, randomization, experimental animals and procedures, and statistical methods.

### Animals

Adult male and female C57BL/6J (B6) and CAST/EiJ (CAST) mice were purchased from Jackson Laboratories (cat. # 000664 and # 000928, respectively; Jackson Laboratories, Bar Harbor, ME) and were housed in the Boston University Laboratory Animal Care Facility (n = 5 per sex for each strain; lights on 7:30 am to 7:30 pm). Mice were euthanized at 8 weeks of age by cervical dislocation, and livers were removed then rinsed in ice cold PBS. For consistency and to minimize the impact of circadian rhythms [118], all mice were euthanized between 10:45 and 11:30 am. Batch effects were minimized for both tissue collection and downstream experiments by including in each group of mice or tissue samples processed, biological replicates from each strain and sex (i.e. male CAST, female CAST, male B6, and female B6 mice). Thus, when collecting livers, no more than two biological replicates for each combination of strain and sex were processed on a given day. A portion of each liver was snap frozen and processed later for RNA isolation (~10% of each liver), or by crosslinking and ChIP-seq (40% of the total mass). The remaining ~50% of each liver was directly used for nuclei isolation (see below).

### RNA isolation and library preparation

RNA was extracted from frozen liver using TRIzol reagent (Invitrogen, cat. # 15596026) following the manufacturer's directions. Total RNA was quantified using the Qubit RNA BR kit

(Invitrogen, cat. # Q10210). One µg of total RNA per sample was used as input for library preparation using the NEBNext Ultra Directional RNA Library Prep kit for Illumina (NEB, cat. # E7420) with the PolyA selection with the NEBNext Poly(A) mRNA Magnetic Isolation Module (NEB, cat. # E7490). Agencourt AMPure XP beads (Beckman Coulter, cat. # A63881) were used to remove low molecular weight material prior to final quantification of purified libraries using the Qubit dsDNA HS kit (Invitrogen, cat. # Q32854).

## Isolation of nuclei, DNase I digestion and library preparation

Nuclei, isolated from freshly excised liver were lightly digested with DNase I to nick accessible chromatin [119], with modifications [120]. Batch effects were minimized by processing 6 samples in parallel but including no more than 2 biological replicates of the same strain and sex in each group of nuclei. Briefly, livers were homogenized in a Potter-Elvehjem homogenizer using high sucrose homogenization buffer (10 mM HEPES (pH 7.5), 25 mM KCl, 1 mM EDTA, 2 M sucrose, 10% glycerol, 0.05 mM DTT, 1 mM PMSF, 0.15 mM spermine, 0.2% (v/v) spermidine, 1 mM Na orthovanadate, 10 mM NaF, and 1X Roche Complete Protease Inhibitor Cocktail, cat. # 11697498001) to prevent aggregation of nuclei and preserve chromatin structure. The resulting slurry was placed on top of a 3 ml cushion of homogenization buffer followed by ultracentrifugation at 25,000 RPM for 30 min at 4˚C in an SW41 Ti rotor to pellet the nuclei and remove cellular debris. The supernatant was carefully decanted to remove liquid, and residual solid debris was removed from the tube walls using a sterile spatula and a dampened Kimwipe. Nuclei were resuspended in 1 ml of Buffer A (15 mM Tris-HCl pH 8.0, 15 mM NaCl, 60 mM KCl, 1 mM EDTA pH 8.0, 0.5 mM EGTA pH 8.0, 0.5 mM spermidine, 0.3 mM spermine) and transferred to a 1.5 ml microcentrifuge tube. Nuclei were taken directly for DNase treatment at this point.

Nuclei were counted using an Invitrogen Countess instrument (cat. # C10281), and 50 million nuclei per liver were incubated in a volume of 950 ul with 32 U of RQ1 RNase-free DNase I (Promega, cat. # M6101) for 2 min at 37˚C. To extract the released DNA fragments, samples were extracted with phenol/chloroform then layered on a sucrose gradient, followed by centrifugation at 25,000 rpm for 24 h at 25˚C. Gradient fractions (1.9 ml each) were collected, and fractions enriched in DNA fragments between 0.1 and 1 kb in size (fractions 7–11; numbered from the top-most fraction) were pooled and then purified by double-sided SPRI bead selection using Agencourt AMPure XP beads to obtain DNA fragments between 125–400 bp long. DNA was quantified using the Qubit dsDNA HS kit and further qualified prior to library preparation by qPCR using primers previously shown to amplify open chromatin regions in mouse liver [36,119]. DNase libraries were prepared for sequencing using NEBNext Ultra II DNA Library Prep Kit for Illumina following the manufacturer's instructions (NEB, cat. # E7645), with double-sided SPRI size selection (Agencourt AMPure XP beads) to obtain an average insert size of 200 bp prior to PCR amplification, and single-sided SPRI after PCR amplification to remove low molecular weight material.

## Chromatin crosslinking, immunoprecipitation, and library preparation

Chromatin was extracted from the ~40% snap-frozen, cross-linked liver aliquots as described [121], with modifications. Batch effects were limited by including no more than 2 biological replicates from the same strain and sex in each set of six liver samples processed together in a single batch. Briefly, livers were resuspended in a total volume of 4.687 ml of XL Buffer (50 mM Hepes, pH 7.5, 100 mM NaCl, 1 mM EDTA, 0.5 mM EGTA, 1X Roche Complete Protease Inhibitor Cocktail), processed with a Dounce homogenizer, and passed through a 70-micron cell strainer (Corning, cat. # 352350). 313 µl of 16% formaldehyde (Pierce, cat. # 28906) was

then added and the samples were nutated for 10 min at 22˚C. Crosslinking was halted by the addition of 250 μl of 2.5 M glycine, followed by additional nutation for 2 min at 22˚C, then incubation on ice for a minimum of 5 min. Cells were pelleted at 2,500 g for 5 min at 4˚C, the supernatant was discarded, and the pellets were rinsed twice with ice cold PBS containing 1X Roche Complete Protease Inhibitor Cocktail, pelleting between each wash as above. The cell pellet was then lysed by resuspension in 10 ml of Lysis Buffer-1 (50 mM Hepes, pH 7.5, 140 mM NaCl, 1 mM EDTA, 10% glycerol, 0.5% NP40, 0.25% Triton X-100) followed by nutation for 10 min at 4˚C. Lysed cells were pelleted, as above, and nuclei were lysed by resuspension in 10 ml Lysis Buffer-2 (10 mM Tris-HCl pH 8.0, 200 mM NaCl, 1 mM EDTA, 0.5 mM EGTA) followed by nutation for 5 min at 4˚C. Crude chromatin extract was pelleted, as above, and resuspended in 2 ml of 0.5% SDS RIPA buffer (50 mM Tris-HCl, pH 8.0, 150 mM NaCl, 1% IPEGAL, 0.5% deoxycholic acid).

Chromatin was transferred to a 15 ml conical tube (Diagenode, cat. # C30010017) for sonication in a Bioruptor Pico instrument (Diagenode, cat. # B01060010). Sonication was performed at 4˚C for 25 cycles (30 s ON and 30 s OFF at high intensity) with 2 min rests on ice every 10 cycles to prevent sample heating. A small aliquot of this material was reverse crosslinked (65˚C for 6 h), treated with RNase A (5 μg per sample at 37˚C for 30 min; Novagen, cat. # 70856) and then digested with proteinase K (20 μg per sample at 56˚C for 2 h; Bioline, cat. # 37084) and electrophoresed on a 1.5% agarose gel to verify that the majority of fragments were 100–400 bp in size. This sonicated material was quantified using PicoGreen Quant-iT dsDNA BR (Invitrogen, cat. # P11496). Prior to antibody incubation (below), ~5% of chromatin by volume was removed and used as a background, pre-immune, input control (total of n = 4 control samples, one per strain per sex).

Immunoprecipitation of mouse liver chromatin was performed as described [52]. Protein A Dynabeads (15 μl; Invitrogen, cat. # 1002D) were incubated in blocking solution (0.5% bovine serum albumin in PBS) with 3 μg of antibody to histone-H3 K27ac (Abcam, cat. # ab4729) for 3 h at 4˚C. Bead immune-complexes were washed with blocking solution, followed by overnight incubation with 15 μg of liver chromatin diluted to a final concentration of 15 ng/μl in RIPA buffer containing 0.1% SDS. After washing with 1X RIPA buffer (containing 0.1% SDS) and reverse crosslinking as described above, DNA was purified using the QIAquick Gel Extraction Kit (Qiagen #28706) and quantified using the Qubit dsDNA HS kit. ChIP-seq libraries were prepared for sequencing using NEBNext Ultra II DNA Library Prep Kit for Illumina, with double-sided SPRI size selection to obtain an average insert size of 200 bp prior to PCR amplification, and single-sided SPRI after PCR amplification to remove low molecular weight material.

## qPCR analysis, pooling of libraries and Illumina sequencing

The quality of all ChIP and DNase-digested samples was validated prior to preparation of sequencing libraries, as follows. qPCR analysis was carried out using primers (Sheet A in S4 Table) for three sex-biased genomic regions, for one sex-independent positive control DHS peak region, for one sex-independent positive control peak region (an H3K27ac-positive site), and for three negative control regions. DNase-qPCR data for sex-biased DHS regions are presented (S5A and S5B Fig) as signal to noise, based on the average of three negative control (background) regions, calculated as follows: $2^{\wedge}[(DHS_{genomic}-DHS_{DNase})/(NegAvg)]$, where $NegAvg = [(NEG1_{genomic}-NEG1_{DNase})+(NEG2_{genomic}-NEG2_{DNase})+(NEG3_{genomic}-NEG1_{DNase})]/3$. "DHS" indicates data obtained with a qPCR primer pair within the DHS region and "NEG" indicates 1 of 3 primer pairs specific to the negative control regions. A subscript "DNase" indicates CT (qPCR cycle threshold) values for qPCR reactions using DNaseI-

digested DNA as template, while a subscript "genomic" indicates $C_T$ values for qPCR reactions using genomic DNA as template. ChIP-qPCR data (S5C Fig) is presented as percent input, calculated as follows: [100/2^(CT value for ChIP DNA–CT value for input DNA)] x [(final volume of ChIP sample) x (initial volume of input sample)] / [(initial volume of ChIP sample) x (final volume of input sample)]. ChIP and DNase-digested sample quality was determined using sex-independent and negative control primers. Samples with the highest signal to noise (Positive Region % input) / (Avg of 3 Neg Ctl region % Input) for ChIP samples; and as described above for DNase-digested samples) were sequenced, as described below.

All sequencing libraries were multiplexed with a dual-indexing strategy using NEBNext Multiplex Oligos for Illumina (NEB, cat. # E7600). Final libraries (n = 5 biological replicates per sex and per strain; 20 libraries each for total liver RNA-seq (polyA-selected), DNase-seq, and H3K27ac ChIP-seq, plus a total of 4 ChIP-seq input control samples (64 libraries in total) were quantified in a single batch using the Qubit dsDNA HS kit. ChIP-seq, RNA-seq, and DNase-seq libraries were pooled on an equimolar basis and sequenced (PE-50 reads) on an Illumina NovaSeq 6000 instrument at the New York Genome Center to an average depth of 34 million read pairs per sample (mean Q $\geq$ 30 for 98.0% of Read-1 and 95.7% of Read-2). We excluded from downstream analysis one of the n = 5 biological replicate DNase-seq libraries per group, due to variable sample quality, as confirmed by qPCR analysis of a sex-independent positive control DHS peak region.

## Custom CAST mouse genome

The B6 mouse genome (mm9) was converted to a custom genome incorporating known genetic variants with g2gtools (v0.2.7; https://github.com/churchill-lab/g2gtools), using quality SNPs/indels downloaded from the Sanger Mouse Genome project, Release 1211 [61]. First, the vcf file from Sanger was converted to an indexed vci file, using the command: "g2gtools vcf2vci —quality -o CAST.vci -s CASTEiJ -p 8 -i mgp.v2.indels.annot.reformat.withchr.vcf.gz -i mgp.v2.snps.annot.reformat.withchr.vcf.gz -f NCBIM37_um_all.fa". Second, SNPs were patched into the mm9 genome using the command: "g2gtools patch -i NCBIM37_um_all.fa -c CAST.vci.gz -o CAST_patched.fa -p 8". Third, Indels were incorporated using the command: "g2gtools transform -i CAST_patched.fa -c CAST.vci.gz -o CAST.fa -p 8". Finally, to annotate the custom CAST genome, custom CAST mouse GTF files were generated for protein-coding gene and lncRNA gene locations converted from mm9 to the newly-generated custom CAST genome using the command "g2gtools convert -i merged_refseq_ncRNA.gtf -c CAST.vci.gz -o merged_refseq_ncRNA.gtf" (for RNA-seq). General conversion of mapped reads, peak lists, or other files with CAST genomic coordinates to B6 (mm9) coordinates was performed using the command: "g2gtools convert -i test_CASTcoord.bed -c CAST.vci.gz -o test_mm9coord.bed -f bed–reverse". CAST-derived sequence reads mapped to the custom CAST mouse genome showed mapping frequencies and cumulative gene expression very similar to B6 mice (Sheet B in S4 Table).

## Mouse liver RNA-seq samples and differential expression analysis

Microarray datasets included in S4E Fig are based on data generated by Jackson Labs and analyzed previously (http://cgd.jax.org/gem/strainsurvey26/v1) [49]. RNA-seq data for CD-1 mice was download from GSE93380 (samples GSM2452246-GSM2452251), and represents three RNA pools per sex, each comprised of polyA-RNA from n = 12 to 17 individual livers per sex [13]. For B6 and CD-1 mouse liver RNA-seq datasets, reads were mapped to the mm9 genome using TopHat (v2.1.1) [122] with default parameters. For CAST mice, sequence reads were mapped to the custom CAST genome (described above) using TopHat. The gene annotation

file used for both strains was identical (prior to coordinate liftover) and included both coding and noncoding genes, as described in [16,123] and implemented in [49]. Mapped reads were filtered to include only primary, uniquely mapping reads, which were assigned to gene-level features based on overlap of at least one nucleotide with any exon using FeatureCounts (v1.6.2) [124]. Reads counts were transformed to fragments per kilobase of exon per million reads mapped (FPKM) for all downstream analysis and visual comparisons. Differentially expressed genes were identified using EdgeR (v3.8) [125] with default parameters. Genes differentially expressed between the sexes were identified based on a male liver vs female liver | fold change| > 2 (i.e., sex bias >2 fold) at an FDR < 0.05 (i.e., EdgeR (Exact test) adjusted p-value < 0.05). We defined strain-unique (strain-specific) sex-biased genes as genes that met the threshold of sex bias > 2 and FDR < 0.05 in either B6 or CAST mouse liver but failed to meet the FDR cutoff in the other strain (FDR > 0.05). Strict strain-shared sex-biased genes met the above sex bias and FDR cutoffs in both strains, while standard strain-shared sex-biased genes met the FDR cutoff in both strains, but the sex bias fold-change cutoff in only one strain. These definitions were without regard to a gene's sex bias in CD-1 mouse liver.

## Mouse liver ChIP-seq and DNase-seq samples and epigenetic data analysis

For B6 mice, ChIP-seq and DNase-seq reads were demultiplexed and mapped to the mouse genome (build mm9) using Bowtie2 (v2.2.9) [126], allowing only uniquely mapped reads. For CAST mice, sequence reads were mapped to the custom CAST genome, allowing only uniquely mapped reads. Peaks of sequencing reads were identified using MACS2 (v2.1.1) [127] as regions of high signal over the background input DNA samples. Peaks were filtered to remove blacklisted genomic regions (www.sites.google.com/site/anshulkundaje/projects/blacklists). Genomic regions called as peaks but that contain only PCR duplicated reads, defined as > 5 identical sequence reads that do not overlap any other reads, were also removed. Differential peaks were identified by diffReps (v1.5.6) [73] with default parameters, then filtered to require a fold-change > 2 (or fold-change > 1.5 for DNase-seq). To account for sample-to-sample variability and increase robustness, we retained for downstream analysis those differential regions that overlap a peak region identified by MACS2, and with a minimum read count of n = 15 for ChIP-seq samples and n = 10 for DNase-seq samples.

Genomic coordinates for reads, peaks, and signal (BigWig) mapped to the custom CAST genome were converted to mouse mm9 coordinates for visualization and normalization. All BigWig tracks used for genome browser visualization were normalized for sequencing depth and fraction of reads in peaks, expressed as reads in peaks per million mapped reads (RIPM) using Deeptools2 (v2.3.3) [128]. Read in peak normalization and diffReps filtering were performed relative to the union of peaks across all samples (B6 and CAST). All browser screenshots are generated using the WashU Epigenome Browser (https://epigenomegateway.wustl.edu). Screenshots presented in the figures are based on samples merged by strain and sex at the fastq level and processed as described above (i.e., all n = 5 male B6 H3K27ac ChIP-seq fastq files were merged; and the same was done for B6 female, CAST male, and CAST female samples). Each dataset was visualized with a pair of tracks representing the signal (*top*; BigWig file normalized as above) and peak/DHS positions (*bottom*; bed file described below). The signal track superimposes merged male (blue) and merged female (pink) replicates (merge of n = 5 for H3K27ac ChIP-seq samples and merge of n = 4 DNase-seq samples; above) in a single track (matplot function of WashU browser). H3K27ac peak and DHS positions are shown for sex-biased peaks and DHS regions (Sheets E-H in S5 Table) and for sex-independent robust peaks and DHS regions (Sheets B and D in S5 Table).

## Proximity of genes to CREs

In Fig 3A-C, distance from the TSS of genes on an indicated gene list (S1 Table) to the nearest CRE on the indicated peak list (S5 Table) was calculated using bedtools (closest -d; bedtools v.2.28.0). For the stacked bar chart in Fig 3D, we assumed that core (strain-shared) sex-biased genes could be regulated by neighboring CREs in one of two ways: 1) sex-biased DHS or K27ac mark accumulation that is sex biased in both strains at the same site (i.e. 533 common CREs; Sheets E and F in S5 Table) or 2) sex-biased DHS or K27ac mark accumulation that is sex biased at a different site in each strain, both of which meet the criteria for the classification (i.e. both a CAST-specific and B6-specific CRE at different sites regulating the same gene; Sheets G and H in S5 Table). For example, a gene was considered proximally-regulated if the TSS of that gene had a CRE within 20 kb that met the criteria described in scenario 1 or 2, above.

## Assigning eQTL and genetic variants to genes

Previous work from this lab identified 1,172 eQTLs associated with the expression of one or more of 1,033 sex-biased genes [49]. For 414 of these eQTLs, the regulating strain in at least one condition is either CAST or B6 (i.e., based on analyses considering either all 438 DO mouse liver samples, 219 male DO liver samples only, or 219 female DO liver samples only). As discussed in [49], for CAST liver, these eQTLs only considered sex-biased genes identified by microarray, which is less sensitive and less specific than RNA-seq and largely limited to protein coding genes ("TRUE" in column AF of Sheet A in S6 Table). To address this issue, we additionally considered any eQTL associated with CAST-specific sex-biased genes (Sheets D and F in S1 Table), which yielded an additional 77 eQTLs ("FALSE" in column AF of Sheet A in S6 Table), giving a final list of 491 eQTLs for which CAST or B6 is the regulating strain associated with 481 sex-biased genes. A separate listing of genes showing sex-biased expression in CAST liver is shown in Sheet C in S6 Table, including annotation for genes not previously identified as sex-biased.

## Definitions for regulating strain and gain or loss of sex bias

The term "regulating strain" is used to indicate the strain in which an eQTL was identified that has the most significant LOD score. The terms "gain" and "loss" of sex bias in the regulating strain are based on the assumption that any private genetic variant (i.e., one unique among all DO founder strains) differs from the ancestral allele.

## Contributing genetic variants

The following strategies were employed to narrow the genomic region under consideration for putative contributing genetic variants. First, for eQTLs significant in >1 set of DO liver samples with B6 or CAST as the regulating strain, we considered up to 3 datasets giving significant eQTLs: all 438 DO liver samples, 219 DO male liver samples only, and 219 DO female liver samples only. We selected the shortest eQTL region of the three (defined as the 95% Bayesian credible interval; [49]). If the eQTL was only significant in B6 or CAST mice in one of these liver sets, then those coordinates were chosen (by default it is the shortest). These values are shown in columns A and B of Sheet A in S6 Table. For eQTLs significant in more than 1 of the 3 sets mentioned above, we determined the intersection of the 2 (or the 3) significant genomic intervals as a secondary set of genomic coordinates, which are listed in columns D and F of Sheet A in S6 Table. Finally, for eQTL intervals that overlap with the TAD containing the TSS of the regulated gene (column Y = "tad"), the intersection of the eQTL and TAD coordinates is

shown in columns G-I of Sheet A in S6 Table. TAD boundaries were those defined for male B6 mouse liver [74,129].

## Discovery of robust sex-biased eQTLs by explicitly modeling sex-QTL interaction

We used the approach of [75] to investigate the hypothesis that eQTLs may have a different effect in each sex. Briefly, the effect of sex on eQTLs was assessed by building two models: Model_A, which includes sex, DO generation, genotyping batch, diet, and sex*diet as additive covariates; and Model_SexInteraction, which in addition to the additive covariates in Model_A, also includes sex as a covariate that interacts with the eQTL (eQTL x sex interaction). The difference of LOD score between the two models (ΔLOD) was used as an estimate of the eQTL x sex interaction, i.e., how different the impact of the eQTL is in each sex. We determined the significance threshold for ΔLOD by performing 1000 permutations of shuffling genotypes and phenotypes for both Model_A and Model_SexInteraction. To ensure that both permutations used the matched permutations of data, we set the seed for the random number generator to be identical before each permutation. The distribution of ΔLOD values was used to determine the genome-wide threshold with α of 0.05 (ΔLOD = 4.27). All the above analyses were performed in R/eQTL2 [130]. The resultant set of 168 robust sex-biased eQTLs regulating sex-biased genes is a subset of the full set of 491 eQTLs regulating sex-biased genes described previously [49], and is identified in Sheet A in S6 Table, columns FS-FU.

## Categorization of eQTLs based on their impact on sex-biased gene expression

The genetic effects uncovered by eQTL analysis in DO mice in relation to the regulating founder strain was described previously [49]. For example, in cases where eQTL analysis in male DO mice yielded an eQTL with a negative regression coefficient, and with CAST as the regulating strain, the gene target of this eQTL is referred to as repressed in CAST male liver. Of the full set of 491 eQTLs regulating sex-biased genes in liver with CAST or B6 as the regulating strain, 355 eQTLs were associated with genes that are significantly sex-biased in B6 and/or CAST mouse liver and 388 eQTLs can be categorized into one of 8 categories described below (Sheets A, column X, and G in S6 Table). The sets of robust sex-specific and strain-specific eQTLs are annotated in Sheet A in S6 Table (columns FU and FV) and in Sheet G in S6 Table (column N). The number of eQTLs that can be categorized (388 of 491) is less than 491 because we only considered eQTLs that are significant in the DO mouse male-only liver set (n = 219), or in the DO mouse female-only liver set (n = 219), and for which CAST or B6 is the regulating strain, or contributes substantially to the regulation as indicated by a regression coefficient within 20% of the absolute maximum regression coefficient in the regulating strain, as determined in [49]. The intersection of the 355 eQTLs regulating genes showing sex-biased expression in either B6 and/or CAST livers with the 388 categorized eQTLs resulted in 286 eQTLs (S9G Fig). eQTLs were categorized based on three factors: the sex-bias of the regulated gene, whether the eQTL was stronger in male or in female liver (by LOD score), and the directionality of the effect on regulated gene expression (positive or negative regression coefficients) (Fig 4C). The sex-bias of the gene was assigned based on the absolute maximum Male/Female log2 fold-change across DO founder strains showing significant sex-biased gene expression (column AJ of Sheet A in S6 Table). Negative values in column AJ of Sheet A in S6 Table indicate female-biased genes (eQTL categories #2, 4, 6 or 8) and positive values indicate male-biased genes (eQTL categories #1, 3, 5 or 7). Next, if the LOD score in DO male livers was higher than in DO female livers (positive value in column AS of Sheet A in S6 Table) then the

eQTL was considered "stronger in male" (categories #1, 2, 5 or 6), and if the LOD score in DO females was higher than in DO males then the eQTL was considered "stronger in female" (categories #3, 4, 7 or 8). Finally, we considered the impact on regulated gene expression as reflected by the regression coefficients in the sex with the stronger eQTL (columns CD-CT and DD-DT of Sheet A in S6 Table). In the sex with the stronger eQTL, a high positive coefficient represents activation of the regulated gene (categories #2, 3, 5 or 8) while a low negative coefficient represents repression of the regulated gene (eQTL categories #1, 4, 6 or 7; Fig 4C).

Impact on gene expression (Fig 4D) was calculated for the 286 eQTLs regulating genes with significant sex-bias in either B6 or CAST mouse liver. For eQTL categories #1–4, a loss of sex-bias was defined as significant sex-bias in the non-regulating strain in combination with no significant sex-bias in the regulating strain (where significant was defined as EdgeR FDR < 0.05; both here and elsewhere in this section). For eQTL categories #5–8, a gain of sex-bias was defined as a significant sex-bias in the regulating strain and no significant sex-bias in the non-regulating strain. For eQTL categories #1–8, a reversal of sex-bias was defined as a significant sex bias in both regulating and non-regulating strains but with opposite directionality. For categories #1–8, sex-bias retained was defined as significant sex-bias in both the regulating and the non-regulating strains, and with the same directionality, but at either a reduced magnitude (eQTL categories #1–4) or an enhanced magnitude (eQTL categories #5–8). The directionality of sex-bias is shown in columns Z (B6) and AA (CAST) of Sheet A in S6 Table, and significance is shown in columns AG (B6) and AH (CAST). As noted above, the terms loss and gain of sex-specific gene expression pattern used here are predictions based on the observed change of sex-specific gene expression pattern in the regulating strain, as compared to other strains examined, and the specific category of eQTL we identified (categories #1–8). However, absent knowledge of the true ancestral allele, we have no way to verify these assignments. Accordingly, a subset of the alleles that we have designated as a gain of sex-specific gene expression could in fact be a loss of sex-specific gene expression pattern in multiple DO founder strains, and vice versa.

Each eQTL was associated with the DO mouse founder strain showing the highest absolute maximum contributing coefficient. An eQTL was said to be associated with more than one DO mouse founder strains if the contributing coefficient to the eQTL for a second strain was within 20% of that of the strain with the highest absolute maximum coefficient. The number of strains contributing to each eQTL is shown in Sheet A in S6 Table, columns BD, CD and DD; and the results are summarized in column FV.

Gain or loss of sex-biased CREs relevant to eQTL regulation (Fig 4E) was calculated for the 388 eQTLs that can be categorized (Sheet A in S6 Table, column X). Relevant CREs were first defined as those that contain strain-specific genetic variants from the regulating strain and that have the same directionality as the regulated gene (i.e., male-biased gene and male-biased enhancer). For eQTL categories #1–4, relevant sex-biased peaks and DHS were defined as those that lose sex-bias in the regulated strain (i.e., for a gene whose male-biased expression is lost in CAST liver, a B6-specific male-biased peak is considered relevant). For eQTL categories #5–8, relevant sex-biased peaks and DHS were defined as those that gain sex bias in the regulated strain (i.e., for a male-biased gene that gains male-biased expression in CAST liver, a CAST-specific male-biased peak is considered relevant). If the eQTL region included only a single relevant CRE (where, for the purposes of counting CREs, overlapping DHS and K27ac peaks are counted only once), then it was designated 'One' in Fig 4E. If multiple relevant CREs fell within the eQTL region, or if no relevant CREs fell within the eQTL, then it was designated 'Multiple' or 'None', respectively (Sheet A in S6 Table, columns DU and DV).

## Strain-specific genetic variants within eQTLs

Both SNPs and Indels were incorporated into the custom CAST genome and were considered as genetic variants. Strain-specific conserved genetic variants were defined as variants that occur in B6 or CAST mice that do not have a genetic variant at the same genomic locus (defined as a single nt in mm9 reference coordinates) in any of the other seven DO founder mouse strains, as described [49]. Strain-specific, non-conserved genetic variants were defined as variants that occur in B6 or CAST mice that do have a genetic variant at the same genomic locus in at least one other DO founder mouse strain, however, the variant is different between the strains (e.g., A -> T in CAST mice and A -> C in PWK mice). Furthermore, only strain-specific conserved and strain-specific non-conserved strain-specific variants were evaluated in this study; we did not consider other genetic variants that distinguish CAST and B6 mice. Genetic variant overlap with CREs was defined using bedtools intersect. Browser screenshots presented in the figures mark only the strain-specific genetic variants within CREs (see S9D Fig). In total, we identified 10,195,846 genetic variants that met our definition of strain-specific in either CAST or B6 mice, of which 124,206 fell within CREs located within eQTL regions (S9D Fig). A B6-specific or CAST-specific variant occurs every 275 bp, on average, calculated based on the size of the mouse genome ($2.8 \times 10^9$ bp) and the total number of strain-specific variants (10,195,846).

The enrichment of strain-specific genetic variants in features (Fig 4B) was calculated for three sets of features: sex-biased CREs, defined as any sex-biased H3K27ac peak or sex-biased DHS in either strain (Sheets E-I in S5 Table); all CREs, defined as any lenient H3K27ac peak or DHS (Sheets A and C in S5 Table), which also includes robust H3K27ac peaks; and intra-motif, defined as any transcription factor (TF) motif, regardless of the expression level or activity of the TF in liver [131]. The database used represents B6 mouse (mm9) genomic coordinates for all predicted mouse TFs identified in [131] by integrating multiple sequence features into a probabilistic Bayesian model. Overlap was defined as a minimum of 1 bp overlap between a genetic variant in the regulating strain with a TF motif. The number of genetic variants within the indicated feature was calculated for all genetic variants genome-wide, all genetic variants within the 491 eQTLs associated with sex-biased genes, and all genetic variants within the intersection of the 383 intra-TAD eQTLs and the TAD containing the TSS of the regulated gene. This number is the observed value. To calculate the expected value, we generated 1,000 size-matched reshuffled features (same number and same width) and calculated the number of genetic variants falling within each reshuffled random feature. The median number for these 1,000 permutations is the expected value. Significance was assessed by empirical p value of the observed overlap relative to the distribution of expected overlaps from the set of 1,000 random permuted regions.

## Data visualization, clustering and general statistical tests

Visualization and clustering of heat maps was performed using Broad's Morpheus software (https://software.broadinstitute.org/morpheus). For visualization in heat maps, gene expression was Z transformed by row. This matrix was then clustered based on average Euclidean distance between each gene across the 20 samples. Principal component analysis was performed using the prcomp and autoplot R packages [132]. Venn diagrams were generated using BioVenn [133]. Aggregate plots and heat maps for ChIP-seq and DNase-seq datasets were generated using Deeptools v2.3.3 [128]. Boxplots, bar graphs, cumulative distribution plots, and scatterplots were generated using GraphPad Prism (v7.0e). Specific statistical tests are noted as used. Generally, nonparametric tests comparing values (i.e., boxplots) used Mann-Whitney (M-W) tests, while distribution comparisons (i.e., cumulative frequency plots)

used Kolmogorov-Smirnov (KS) tests. For graphs, p-values are marked by asterisks, as follows: *, $p < 0.05$; **, $p < 0.01$; ***, $p < 0.001$; and ns (not significant), $p > 0.05$. Boxplots are graphed to show 1.5*IQR (interquartile range, Tukey) and bars plots show the SEM around the mean per group.

### Gene Ontology and functional enrichment analysis

Functional annotation clustering for all indicated gene lists was performed using DAVID v6.8 [134] with high stringency for clustering, but otherwise using default parameters. Gene Ontology enrichment for molecular function was performed using DAVID using only the field "GOTERM_MF_DIRECT" and the output was ranked on the FDR column. Strain-specific, sex-biased TFs were identified from the list of 301 B6-unique or 207 CAST-unique sex specific gene lists. Strain-shared TFs were identified from the combined list of 211 standard and strict common sex-biased genes, which was filtered based on the associated gene ontology terms using DAVID analysis, based on molecular functions related to DNA binding or transcription factor activity (GO:0003677, GO:0043565, or GO:0000981). For shared sex-biased TFs to be considered expressed, a minimum of 2 of the 6 subgroups (male and female B6, CAST, and CD1 liver) had to have a median expression $\geq 1$ FPKM. For a strain-specific TF to be considered expressed, the median expression in that strain must be $\geq 1$ FPKM for at least one sex.

### Supporting information

**S1 Fig. Expression data across strains and sexes for select strain-conserved sex-biased genes and PCA analysis. A-D**. The top of each panel shows aggregated Z scores for cluster MA and FA indicated in Fig 1B (for protein-coding genes) and 1C (for lncRNA genes). The bottom panels show mouse liver expression data for the 4 indicated strain-conserved sex-biased genes, in FPKM units for 6 pooled CD-1 samples (n = 3 per sex), 10 individual B6 samples (n = 5 per sex), and 10 individual CAST samples (n = 5 per sex). **C, D**. Principal component analysis (PCA) for the 144 strain shared sex-biased protein coding genes (C) and for the 78 strain shared sex-biased lncRNA genes (D) reveals separation by sex along PC1 and by strain along PC2. The variance along each principal component is shown on each axis. (PDF)

**S2 Fig. RNA-seq data for strain-unique sex-biased protein coding genes.** Heat maps presenting relative expression levels across individual mouse livers (n = 20; 5 per sex in each strain) for 301 B6-unique sex-biased protein-coding genes (**A**) and for 207 CAST unique sex-biased protein-coding genes (**B**), based on data in Sheets C and D in S1 Table. Expression values are shown as Z-scores normalized per row to visualize patterns independent of the expression level. Log2 (M/F fold-change) values are shown at the left, with blue indicating male bias and purple indicating female bias for B6 (column marked 'B') and CAST datasets (column marked 'C'). Hierarchical clustering was performed based on Euclidean distance and is shown to the left of each heat map, with colors indicating the cluster identity labelled to the right of the heat map. In **A**, clusters B1 and B2 comprise 238 genes that show higher expression in B6 mouse liver; and in **B**, clusters C1 and C2 comprise 142 genes that show higher expression in CAST mouse liver. (PDF)

**S3 Fig. RNA-seq data for strain-unique sex-biased lncRNA genes.** Heat maps presenting relative expression levels across individual mouse livers (n = 20; 5 per sex in each strain) for 289 B6-unique sex-biased lncRNAs (**A**) and for 187 CAST-unique sex-biased lncRNAs (**B**), based on data in Sheets E and F in S1 Table. Formatting and presentation are as described in S2 Fig.

In **A**, clusters B1 and B2 comprise 253 genes that show higher expression in B6 mouse liver, and in **B**, clusters C1, C2, C4, C5 comprise 152 genes that show higher expression in CAST mouse liver. The reduced number of CAST-specific sex-biased lncRNAs could be a reflection of more rapid evolution of non-coding genes, but also is likely a due to the inclusion of B6 but not CAST mice in the datasets used for discovery and annotation of liver-expressed lncRNA genes [1, 2].
(PDF)

**S4 Fig. Expression data for TFs related to sex bias that do not show strain-specific expression.** Expression data are presented as FPKM values for 6 pooled CD-1 livers (n = 3 per sex), 10 individual B6 livers (n = 5 per sex), and 10 individual CAST livers (n = 5 per sex). **A.** Three transcriptional regulators with well-established roles in liver sexual dimorphism (Stat5b, Foxa2, and Hnf4a). No sex-dependence or strain bias is found. **B.** Three transcriptional regulators that are expressed in a sex-dependent manner, and in the case of Cux2 and Bcl6, have established roles in the regulation of liver sexual dimorphism. **C**. Examples of sex-biased TF genes from S2A Fig, clusters B2, B3 and B4. These genes are putative transcriptional regulators presented in S3 Table. The three B6-unique sex-biased TFs also show sex-biased expression in CD-1 livers (fold-change > 1.5 and FDR < 0.05). **D.** Examples of sex-biased TF genes from S2B Fig, clusters C3 and C5. **E.** Shown is the total number of liver sex-biased protein-coding (PC) genes (*top*) and lncRNA genes (*bottom*) across B6, CAST, and CD-1 mice. The impact of increasing the threshold for sex bias from standard (2-fold) to 4-fold, both at FDR< 0.05, is also shown. From *left* to *right*, the number of sex-biased genes at the specified threshold is shown for the following groups: PolyA+ RNA-seq from B6 and CAST (this study), and from CD-1 mouse liver [3], B6 microarray (Jax Strain Survey 26), and CAST microarray (Jax Strain Survey 26; http://cgd.jax.org/gem/strainsurvey26/v1). A total of 852, 661, and 826 standard sex-biased genes were identified in B6, CAST, and CD-1 mice, respectively (protein coding and lncRNA genes combined). Similarly, a total of 391, 313, and 389 strictly sex-biased genes were identified in each strain, respectively.
(PDF)

**S5 Fig. Strain-dependent epigenetic marks nearby sex-biased genes identified by DNase-qPCR and H3K27ac ChIP-qPCR.** We used qPCR to interrogated known sex-biased enhancers, identified in CD-1 mouse liver (browser screen shots to the *right* of each panel), within 15 kb of the female-biased gene *Cyp3a16* (**A**), and the male-biased genes *Cyp7b1* (**B**) and *Gstp1* (**C**). All three genes have nearby CAST-specific TF binding sites, as identified in male mouse liver [4]; these strain-specific sites are statistically significant in all cases but with a magnitude of strain-preference varies, ranging up to >100-fold CAST-specific, as marked in the panels on the right side of each figure. Expression values (in FPKM) are from S1 Table and are based on RNA-seq using male B6 (cluster MB; n = 5), female B6 (cluster FB; n = 5), male CAST (cluster MC; n = 5), and female CAST (cluster FC; n = 5) mouse liver. The magnitude of sex bias calculated by EdgeR is shown above each pair of gray and green bars (linear M/F values). Genomic locations of the qPCR amplicons used to interrogate the DHS and H3K27ac-ChIp'd DNA are indicated in each browser panel, with primer sequences shown in Sheet A in S4 Table. Strain-specific TF binding sites are listed in S6 Table. DNase-qPCR results (set of 4 bars at the middle of each panel) are presented as the signal in the indicated genomic region divided by the average of 3 negative control regions (see Methods). **A.** Female-biased chromatin opening for a DHS on chromosome 5 near *Cyp3a16* is seen in CD-1 and CAST but not B6 mouse liver. For the enhancer neighboring *Cyp3a16*, female-biased chromatin opening is seen in CAST mouse liver (*, p = 0.0159) but not B6 mouse liver. Further, greater chromatin opening is seen in CAST relative to B6 (male B6 vs male CAST, * p = 0.0168; and female B6 vs female

CAST liver, ** p = 0.0072; t-test). This strain difference in chromatin opening could be explained by CAST-specific binding of the TF Foxa1 at this genomic region, even though *Cyp3a16* is not differentially expressed between the strains. The WashU Epigenome Browser screenshot (*right*) shows the DNase-seq signal for male (blue) and female (pink) CD-1 liver for the genomic region upstream of *Cyp3a16*, which includes four female-biased DHS (pink bars underneath DHS track). The female-biased DHS interrogated by qPCR overlaps a CAST-specific Foxa1 peak with 7.4-fold CAST binding preference (green horizontal bar; [4]). **B**. Male-biased chromatin opening for a DHS enhancer ~11 kb upstream of the male-biased gene *Cyp7b1* is seen in CD-1 and B6 but not in CAST mouse liver. Correspondingly, *Cyp7b1* expression is much higher in CAST than in B6 female mouse liver, and the sex bias in expression is much lower in CAST liver. Browser screen shot shows the DNase-seq signal intensity for male and female CD-1 liver for the region immediately downstream of *Cyp7b1*, including 4 male-biased DHS (blue horizontal bars). DNase-qPCR results indicate a male-biased DHS in B6 but not CAST liver. This DHS overlaps a CAST-specific Cebpa peak with 1.5-fold CAST liver binding preference (green horizontal bars; [4]). **C**. ChIP-qPCR for genomic regions enriched for acetylation of lysine 27 on histone 3 (H3K27ac), which is associated with active promoters and enhancers. ChIP-qPCR results indicate male-biased H3K27ac marks in both mouse strains at an enhancer ~10 kb upstream of *Gstp1*, although the magnitude of male-biased gene expression is higher in B6 mouse liver. Browser screenshot shows H3K27ac ChIP-seq signal for male and female CD-1 liver for the region immediately upstream of *Gstp1*. A strong CAST-specific Foxa1 peak with >100-fold CAST binding preference (green horizontal bar; [4]) overlaps a separate male-biased H3K27ac peak.
(PDF)

**S6 Fig. Differential ChIP-seq and DNase-seq results. A.** Number of differential H3K27ac peaks between male and female B6 mouse liver (y-axis) binned by the magnitude of fold-change in sex bias (log2 F/M as calculated by diffReps; x-axis). Shown are those sites that overlap MACS2 peaks. Blue bars indicate male-biased sites, red bars indicate female-biased sites, black bars indicate sites below the 2-fold fold-change cutoff (either direction), and gray bars indicate sites with low read count (minimum 15 reads in up regulated condition). In B6 mouse liver, differential analysis of in-peak diffReps sites identified 2,288 sex-biased H3K27ac peaks, of which 1,256 were male-biased and 1,032 were female-biased. To be considered sex-biased, the M/F or F/M fold-change > 2 with an FDR < 0.05 (n = 5 per sex; see Methods). **B.** Shown is the number of differential H3K27ac sites between male and female CAST mouse liver, as described for panel A. In CAST mouse liver, differential analysis of in-peak diffReps sites identified 2,571 sex-biased H3K27ac peaks, of which 795 were male-biased and 1,776 were female-biased. To be considered sex-biased, the M/F or F/M fold-change > 2 with an FDR < 0.05 (n = 5 per sex). Numbers differ slightly from Fig 2A due to small number of sites showing divergent sex bias between strains and in some cases a peak region identified in one strain that overlaps two peaks in one of the other strains in Fig 2A. **C.** Shown is the number of differential DHS between male and female B6 mouse liver (y-axis) binned by the magnitude of fold change sex bias (log2 F/M as calculated by diffReps; x-axis). Shown are those sites that overlap MACS2 peaks. Blue bars indicate male-biased sites, red bars indicate female-biased sites, black bars indicate sites below the 1.5-fold fold-change cutoff (either direction), and gray bars indicate sites with low read count (minimum 10 reads in upregulated condition). In B6 mouse liver, differential analysis of in-peak diffReps sites identified 1,311 sex-biased DHS, of which 400 were male-biased and 911 were female-biased. To be considered sex-biased, the M/F or F/M fold-change > 1.5 with an FDR < 0.05 (n = 4 per sex). **D.** Shown is the number of differential DHS between male and female CAST mouse liver as described for panel C. In CAST mouse

liver, differential analysis of in-peak diffReps sites identified 986 sex-biased DHS, of which 152 were male-biased and 834 were female-biased. To be considered sex-biased, the M/F or F/M fold-change > 1.5 with an FDR < 0.05 (n = 4 per sex). Numbers differ slightly from Fig 2B, as explained in panel B. **E.** Core sex-biased enhancers are consistently sex biased between individual biological replicate livers and across strains. Normalized ChIP-seq and DNase-seq signal are displayed around the peak midpoint +/-10 kb for 441 shared sex-biased ΔK27ac peaks (*top*) and 92 shared sex-biased ΔDHS (*bottom*). At right is shown aggregate plots for the indicated groups: male-biased H3K27ac peaks, female-biased H3K27ac peaks, male-biased DHS, and female-biased DHS. Groupings are indicated below the heat maps for males (horizontal blue bar), females (horizontal pink bar), B6 mice (gray bar), CAST mice (green bar). Aggregate plots for the indicated groupings (*right*) are shown for individual B6 males (dark blue), CAST males (light blue), B6 females (dark pink), and CAST females (light pink). **F.** All protein coding and lncRNA TSS were used for this analysis. Peak groups from *left* to *right* are (1) all lenient K27ac peaks or DHS (Sheets A and C in S5 Table), (2) all robust K27ac peaks of DHS (Sheets B and D in S5 Table), (3) "core" sex-biased K27ac peaks or DHS (Sheets E and F in S5 Table), (4) B6-unique sex-biased K27ac peaks or DHS (Sheets G and H in S5 Table), and (5) CAST-unique sex-biased K27ac peaks or DHS (Sheets I and J in S5 Table). **G.** Shown are aggregate plots of the conservation score for a 5-kb window surrounding the peak or the DHS midpoint for strain-shared peaks, strain-unique peaks, strain-shared sex-biased peaks, and strain-unique sex-biased peaks. Conservation scores representing vertebrate conservation (PhastCons 30-way; comparing 30 vertebrate species) were calculated for lenient strain-shared and lenient strain-unique peaks, as well as the strain-shared sex-biased peaks ("core") or strain-unique sex-biased peaks (S4 Table). Scores are only shown for regions not overlapping coding regions, which are considered empty (nonzero) values.
(PDF)

**S7 Fig. Additional core male-biased genes with local core male-biased enhancers.** Annotations and formatting are as described in Fig 5. **A.** *Gstp1* is a male-biased gene in both B6 and CAST mouse liver (M/F fold-change = 9.2 and 5.3, respectively). **B.** *Elovl3* is a male-biased gene in both B6 and CAST mouse liver (M/F fold-change = 59.7 and 55.7, respectively). **C.** *Nudt7* is a male-biased gene in both B6 and CAST mouse liver (M/F fold-change = 3.5 and 4.0, respectively).
(PDF)

**S8 Fig. Additional core female-biased genes with local core female-biased enhancers.** Annotations and formatting are as described in Fig 5. **A.** *Acot3* and *Acot4* are female-biased genes in both B6 and CAST mouse liver. *Acot3* shows a greater magnitude of female-biased expression (F/M fold-change = 34 in B6 and 69 in CAST) compared to *Acot4* (F/M fold-change = 3.5 in B6 and 4.9 in CAST). **B.** *Cyp4a14* is a female-biased gene in both B6 and CAST mouse liver (F/M fold-change = 18.4 and 416, respectively). **C.** *Cux2* is a female-biased gene in both B6 and CAST mouse liver (F/M fold-change = 239 and 416, respectively).
(PDF)

**S9 Fig. Approach for considering only CAST and B6 founder strains and filtering all DO mouse eQTLs for relevant variants within CREs. A.** 491 eQTLs have CAST or B6 as the regulating strain in at least 1 of 3 comparisons: all DO mouse liver samples, DO male liver samples only, and DO female liver samples only. We identified 406 eQTLs for which CAST is the regulating strain and 85 for which B6 is the regulating strain. There are 7 possible combinations based on significant in all liver samples (A), significant in male liver samples (M), and significant in female liver samples (F), as indicated below the x-axis. The distribution of eQTLs in

each of these 7 groups (per strain) is shown. **B.** *Left*: Boxplots indicating the absolute Log2 fold change of genes regulated by lenient eQTLs (blue) and genes regulated by robust eQTLs (red). The absolute fold change (M/F or F/M) of genes regulated by robust eQTLs is higher than those of genes regulated by lenient eQTLs (p = 6.12e-17, student's T-test). *Right*: Boxplots indicating the maximum expression in Log2(FPKM+1) of genes regulated by lenient eQTLs (blue) and genes regulated by robust eQTLs (red). The median expression of genes regulated by robust eQTLs is higher, but this difference is not significant (p = 0.0831, student's T-test). **C.** *Left*: Boxplots indicating the distribution of sizes of eQTLs for which B6 (median size 2.14 Mb; n = 406) or CAST (median size 1.7 Mb; n = 85) is the regulating strain. *Right*: Boxplots indicating the number of strain-specific variants in eQTLs for which B6 (median variant count 320; n = 85 eQTLs) or CAST (median variant count 6,299; n = 406 eQTLs) is the regulating strain. **D.** Flowchart depicting the filtering strategy for identifying strain-specific SNPs and Indels relevant for sex-biased gene expression. All variants specific to CAST or B6 mice across the 8 founder strains were filtered to consider only those that fall within the 491 eQTLs associated with sex-biased genes, and for which CAST or B6 is the regulating strain. Finally, variants falling within the TAD-eQTL overlap and within cis-regulatory elements are considered as the most likely candidate regulatory variants. Functionally, this approach filters the total of 10,195,846 strain specific variants down to 32,050 intra-TAD and intra-eQTL variants that may be associated with gain or loss of sex bias. **E.** Diagram of the genomic regions considered, based on either: 1) the entire eQTL span (blue); 2) the TAD that contains the TSS of the regulated gene (orange); or 3) the intersection of the TAD with the eQTL containing the TSS (green). **F.** Impact of considering the intersection of the multiple overlapping eQTLs, or the intersection of the eQTL with the TAD containing the TSS of the regulated gene, on interval size and number of strain-specific variants in the regulating strain per interval. *Top*: Boxplots showing the impact on interval size in kilobases. *Bottom*: boxplots showing the impact on number of variants per interval. Intersection of intervals is only applicable to the n = 218 eQTLs that are significant in at least 2 of 3 groups described in S9A Fig (A, M, and F) and are overlapping. TAD-eQTL intersection is only possible for the n = 383 intra-TAD eQTLs (Fig 4A). See Sheet A in S6 Table, columns Q-S (interval size) and columns T-V (number of strain-specific variants). **G, H.** Shown in **G** is a Venn diagram indicating the 286 eQTLs to which we were able to assign "gain/enhancement" or "loss/reduction" of sex-bias, as shown in Fig 4C. Of the 491 eQTLs regulating sex-biased genes that also have CAST or B6 as the regulating strain, 355 regulate genes that are sex-biased in either CAST or B6 mouse liver, as determined in this study by differential analysis of RNA-seq datasets. Similarly, 388 of the 491 eQTLs could be grouped into categories #1–8 (Fig 4C) based on stronger eQTL effect in male or in female DO mouse liver, with CAST or B6 as the regulating strain in the set of DO mice with the stronger LOD effect (see Methods). 286 eQTLs represent the overlap between these two groups, which are significantly sex biased in either B6 or CAST mouse liver (FDR < 0.05; no fold-change restriction) and are significant in DO male liver alone or DO female liver alone for categorization. Further details on eQTL characterization are shown in **H**.
(PDF)

**S10 Fig. Single causal variants within eQTLs related to strain-specific sex-biased gene expression.** Annotations and formatting are as described in Fig 5. **A.** The male-biased gene *Olfm2* is repressed in CAST male liver, resulting in a loss of sex-biased expression in CAST mice (category #1 eQTL). This repression is associated with multiple genetic variants falling within a male-biased peak that is lost in CAST mice (this is the only such region in the eQTL; see Sheet A in S6 Table). Two of these variants show substantial binding preference for B6 over CAST for the TFs Cebpa (16-fold) and Foxa1 (9-fold) in male mouse liver (orange bars).

CAST is the regulating strain for this gene only in male DO mice (LOD = 10.41 in DO males with regression coefficient of -1.0). Significant male-biased expression is observed only in B6 (11-fold M/F) and not in CAST mouse liver. **B.** The female-biased gene *Enpp1* is activated in B6 female liver, resulting in a gain of sex-biased expression in B6 mice (category #8 eQTL). This repression is associated with 4 potential genetic variants falling within a female-biased peak that is only female-biased in B6 (this is the only such region in the eQTL; see Sheet A in S6 Table). B6 is the regulating strain for this gene only in females and no strain is significant in male DO samples (LOD = 10.44 in CAST female with regression coefficient of -0.91) Significant female-biased expression is observed only in B6 (2.1-fold F/M) and not in CAST liver. **C.** The male-biased gene *Bok* is repressed in CAST female liver, resulting in a small increase in the magnitude of sex-bias in CAST liver (category #7 eQTL). This repression is associated with multiple genetic variants within a *Bok* intronic H3K27ac enhancer peak that is male-biased in CAST but is not sex-biased in B6 liver. CAST is the regulating strain for this gene only in female DO mice (LOD = 10.44 in DO females; regression coefficient, -0.91). The significant male-biased expression seen in B6 liver (2-fold M/F) is retained, and is moderately increased in CAST liver (2.3-fold F/M). Browser screenshot shows a CRE with male-biased H3K27ac in CAST but not B6 liver. Data presented as in Fig 5. **D.** Browser screenshot shows normalized sequence read signal tracks, as in panel C, but superimposed by sex. This allows direct comparison of normalized sequence read abundance between B6 and CAST mouse liver of the same sex. The lower H3K27ac mark accumulation at the promoter of *Bok* in CAST female compared to B6 female liver (second track from the top) accounts for the observed gain of male bias at this CRE (a characteristic of category #7 eQTL). B6 tracings are shown in pink (female) or blue (male), while CAST tracings are shown in green for all tracks. The lower expression of *Bok* in CAST male liver compared to B6 male liver, seen in the expression data in panel A, could be due to the decrease in chromatin opening (DNase-seq data) that is seen in CAST male liver (asterisk).
(PDF)

**S11 Fig. Additional example of coordinated activation of multiple enhancers in female leading to increased sex-biased gene expression (*Bmper*, category #8 eQTL).** Annotations and formatting are as described for Fig 5. **A.** The female-biased gene *Bmper* is activated only in CAST female liver, resulting in a robust enhancement of female-biased expression in CAST mouse liver (significant only in CAST female; category #8 eQTL). This activation is associated with multiple genetic variants falling within several CREs, including both strain-shared male-biased peaks and CAST-unique male-biased peaks. CAST is the regulating strain only in female DO mouse liver, suggesting a strong sex-dependence of this effect (LOD = 18.69 in DO male livers with regression coefficient of +1.19). Given that there are two strain-shared (or "core") female-biased enhancers, it is reasonable to conclude that one or both are sufficient to maintain female-biased expression (4.9-fold F/M in B6), while the additional sites are either additive or synergetic in further enhancing the expression and female bias in CAST liver (F/M fold-change of 17.4 in CAST). **B.** Shown is a WashU Epigenome Browser screenshot containing multiple CRE elements that gain female bias and activity in CAST liver in the genomic region neighboring the female-biased gene *Bmper*. Gray bars indicate single variants and red indicates multiple variants (2 or more) in the CAST Variants track. Additionally, three tracks indicate the strain-specificity of transcription factor binding (green indicates CAST-specific; orange indicates B6-preference) for three TFs: Cebpa, Foxa1, and Hnf4a (top to bottom).
(PDF)

**S12 Fig. Coordinated CRE activation in female CAST liver leads to loss of male bias. A.** Three genes showing significant male-biased expression in B6 but not CAST liver. All three

genes are up regulated by category #3 eQTLs, with CAST being the regulating strain in female but not male DO liver (S12 Fig), resulting in the loss of sex biased expression in CAST liver. *, significant M/F expression at FDR < 0.05. Data is presented as described in Fig 5. **B.** Browser screenshot showing multiple CREs male-biased in B6 but not CAST liver; only one CRE retains partial male bias in CAST liver (black asterisk). CAST Variants tracks: gray bars, single variants; red bars, multiple variants. Unique: strain-specific variants; All, all variants between B6 and CAST liver. Three other tracks indicate strain-specificity for transcription factor binding: green, CAST-preferential; orange, B6-preferential, as in Fig 7B. Red asterisk: DHS showing the relevant pattern of sex bias (loss of male bias in CAST) that lacks a strain-specific SNP/Indel. Red arrows, two CREs (K27ac and DHS) that show strain-specific sex dependence While the DHS at the TSS of *lnc9349* does not contain a CAST-specific SNP/Indel, it does contain two SNPs that are shared between CAST and PWK, which are different from the mm9 reference allele (B6). Data is presented as described in Fig 5. **C, D, E.** eQTL analysis for three B6 mouse liver male-biased genes that loose sex-specific expression in CAST mouse liver. *Rassf3*, *lncRNA9349* and *lncRNA9351* all show male-biased expression in B6 mouse liver that is lost in CAST mice due to an activating eQTL seen in female, but not male DO mice (category #3: activation of a male-biased gene in female liver; Fig 4C), for which CAST was identified as the regulating strain, as seen here. No regulating strains were identified for male DO mice for *lnc9349*, and neither B6 nor CAST were the regulating strains for *Rassf3* or *lnc9351*. Annotations and formatting are as described for Fig 5.
(PDF)

**S13 Fig. eQTL analysis and browser screenshots for *Sult3a-Rsph4a* region on chr10.** Annotations and formatting are as described for Fig 5. **A-C.** eQTL analysis results for three sex-biased genes from chr10, with data presented as in Fig 5. Extensive strain effects are seen in both male and female for *Gm4794* and *Sult3a2* (positive coefficients in male and female DO mice with CAST as regulating strain, but stronger in male liver), whereas *Rsph4a* shows significant negative regulation in female but not male liver with CAST as regulating strain. **D.** Zoomed in screenshot for the *Sult3a1* region (gene body and ~10 kb upstream) shown in Fig 8B, to better visualize the female-biased enhancer activity unique to CAST mice. Red asterisks indicate CREs showing strain bias but that lack any strain-specific SNPs/Indels (see Discussion). **E.** Zoomed in screenshot for the *Rsph4a* promoter region shown in Fig 8B, to better visualize the female-biased promoter activity unique to B6 mice. Of several CAST-specific variants within the B6-specific female-biased H3K27ac peak, one variant disrupts an HNF6 binding motif. This is also the only HNF6 motif within a female-biased HNF6 binding site in CD-1 liver in this eQTL region. We hypothesize that disruption of this motif results in observed loss of female-biased expression in CAST mouse liver seen in Fig 8A.
(PDF)

**S1 Table. Strain-shared and strain-unique (strain-specific) sex-biased genes.** Included are genes that are sex-biased in both B6 and CAST (sheets A and B), and genes that are sex-biased only in B6 (sheet C for PCGs, sheet E for lncRNAs) or only in CAST (sheet D for PCGs, sheet F for lncRNAs).
(XLSX)

**S2 Table. Gene Ontology analysis.** Shown are the results of GO term analysis (GO molecular function and functional annotation clustering) for core sex-biased genes and strain-unique sex-biased genes.
(XLSX)

**S3 Table. Sex-biased transcription factors.** Sheet A shows B6/CAST shared sex-specific TFs, as described above. Sheet B shows putative B6-unique sex-specific TFs. Sheet C shows putative CAST-unique sex-specific TFs.
(XLSX)

**S4 Table. Primers, RNA-seq, ChIP-seq, and DHS-seq stats.** Summary of quality control metrics on a per sample basis and those used to qualify samples. This includes primer sequences used for qPCR in sheet S4A. Five primer pairs were used to determine signal over noise for two sex independent regions and 3 negative control regions. RNA-seq, ChIP-seq, and DNase-seq metrics are presented in sheets 4B, 4C, and 4D, respectively.
(XLSX)

**S5 Table. ChIP-seq and DHS peak regions.** Sheets A-D shows the genomic coordinates of all peaks and DHS, including: all K27ac peaks, all robust H3K27ac peaks, all DHS, and all robust DHS. Sheets E and F show the genomic coordinates of core sex-biased H3K27ac peaks and core sex-biased DHS. Sheets G-H show the genomic coordinates of strain unique sex-biased peaks and DHS, including: B6-unique sex-biased peaks/DHS and CAST unique sex-biased peaks/DHS.
(XLSX)

**S6 Table. eQTL listings.** Summary of eQTL analysis, incorporating findings from RNA-seq, ChIP-seq, and DNase-seq analysis of livers from male and female B6 and CAST mice. The primary results are presented in sheet A, summarizing characteristics of the full set of 491 eQTLs, and identifies the robust subset of 168 eQTLs described in the text. Sheet B lists the peak names of all peaks overlapping variants within eQTLs, which can be cross-referenced with S5 Table. Sheet C lists the sex-biased genes unique to CAST that were used to supplement the sex-biased genes identified previously by Melia and Waxman (only CAST microarray data was available previously). Sheets D-F provide strain-specific and strain-shared TF binding information for Cebpa, Foxa1, and Hnf4a. Sheet G identifies key eQTL subsets described in the text.
(XLSX)

## Author Contributions

**Conceptualization:** Bryan J. Matthews, David J. Waxman.

**Data curation:** Bryan J. Matthews.

**Formal analysis:** Bryan J. Matthews, Tisha Melia.

**Funding acquisition:** David J. Waxman.

**Investigation:** Bryan J. Matthews, David J. Waxman.

**Methodology:** Bryan J. Matthews, Tisha Melia, David J. Waxman.

**Project administration:** David J. Waxman.

**Resources:** Bryan J. Matthews.

**Software:** Bryan J. Matthews, Tisha Melia.

**Supervision:** David J. Waxman.

**Validation:** Bryan J. Matthews.

**Visualization:** Bryan J. Matthews, David J. Waxman.

**Writing – original draft:** Bryan J. Matthews, David J. Waxman.

**Writing – review & editing:** David J. Waxman.

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
