## [Decision Letter · Decision Letter 0]

21 Jun 2021

Dear Dr Waxman,

Thank you very much for submitting your Research Article entitled 'Harnessing natural variation to identify cis regulators of sex-biased gene expression in a multi-strain mouse liver model' to PLOS Genetics.

The manuscript was fully evaluated at the editorial level and by independent peer reviewers. The reviewers appreciated the attention to an important topic but identified some concerns that we ask you address in a revised manuscript. In particular, both reviewers questioned why only strain-specific variants were included in the analyses and not all variants that segregate between the two strains. There is also a concern about performing separate analyses for males and females and the ability to make comparative statements about these results across sexes. More detailed descriptions about the methodology of some of the analyses is requested.  Lastly, both had questions about the strength of the evidence for causality, with recommendations for how this analysis could be strengthened.  

We therefore ask you to modify the manuscript according to the review recommendations. Your revisions should address the specific points made by each reviewer.

[LINK]

Yours sincerely,

Terry Furey, PhD

Guest Editor

PLOS Genetics

John Greally

Section Editor: Epigenetics

PLOS Genetics

Reviewer's Responses to Questions

**Comments to the Authors:**

Reviewer #1: In their manuscript “Harnessing natural variation to identify cis regulators of sex-biased gene expression in a multi-strain mouse liver model”, Matthews and Waxman identify potentially causal relationships between eQTLs, sex-biased CREs, and sex-biased gene expression in B6 and CAST mice. This is an extension of their previous work identifying eQTLs for sex-biased protein coding genes and lncRNAs. Here, they identify a core group of strain-conserved sex-biased genes that are proximally regulated, have greater sex bias, and are related to liver function. In comparison to conservation of sex-biased gene expression, the fraction of sex-biased CREs conserved across strains is much lower and there is little sequence conservation between those that are strain-shared, indicating that the mechanism of gene expression regulation is not explained by sequence alone and may differ by strain. This work provides insight into epigenetic regulation of sex-biased gene that can be easily extended in translational work to uncover sex-specific responses to liver disease and treatments in humans.

There is one concern regarding the statistical methods used for the eQTL analysis. The identification of eQTLs in female and male mice is done by separate analysis of the samples by sex. Doing so reduces power and increases the chances that there observed sex differences are due to type II error. Instead, a sex-QTL interaction term should be included in a model using the full data.

Another shortcoming of the analysis is the filtering of genetic variants to those that are strain-specific. This limitation was addressed by the authors in the discussion (line 552-554), but relaxing this constraint to include any variants that differ between B6 and CAST would provide a more complete picture of potential causal regulators and help clarify cases where no strain-specific variant was identified within the CRE.

The examples illustrating the gain or loss of a single sex-biased CRE provide compelling biological evidence of a causal relationship between the eQTL, CRE, and gene expression. For these examples, regressing gene expression on the eQTL while conditioning on the nearby H3K27ac peak and/or DHS could provide further evidence that the mechanism by which the eQTL is influencing gene expression is epigenetic modifications at the CRE. Doing the same for examples where there are multiple sex-biased CREs may also elucidate which CREs are regulators of sex-biased gene expression, and which may be reacting independently to the QTL.

Minor Concerns:

• There is no color in the legend for figure B and C

• It is best practice to show differences in means with points and error bars instead of bar charts with error bars (Krzywinski and Altman, 2014). Thus, Figure S5A-C should show points with error bars instead of the bars.

• Replace C57Bl/6J with C57BL/6J throughout.

Krzywinski, M., Altman, N. Visualizing samples with box plots. Nat Methods 11, 119–120 (2014). https://doi.org/10.1038/nmeth.2813

Reviewer #2: This is an interesting paper that carefully analyzes a rich data set to better understand the interactions between sex, genetic variants, and gene regulation in the liver. I’ve included a number of suggestions and a few concerns. Chief among them is point #19 – the absence of some SNPs that segregate between the two strains under study fundamentally changes the results. The second concern is implied causality, which could largely be addressed through softening some language or moving interpretation to the discussion section.

Intro

1. Line 129 – some will quibble with the purity of the inbred strain subspecies assignment. The best estimates are in the 90+% range. See Yang 2011. The authors might simply write “mostly M.m.domesticus” instead.

Section 1

3. What is the analysis for the statement “showing the same expression pattern” (line 158)? For example, what separates FA, FB2, and FC1? There are no methods described for cluster assignments.

4. The heatmap suggests there is higher variance among replicates for FB2 – is there a way to quantify that?

5. Cd-1 is included inconsistently – present in Fig1A, absent in Fig 1B/C. The text mirrors this – Cd-1 mentioned in only one sentence but since it is the second sentence of the result it sets expectations for some treatment of those results. For instance, is there higher sharing between B6 and CD-1 because of subspecies similarity?

6. “Strain-shared” was confusing the first time I read it (partly because there are three strains). In Results Section 3 you switch to “strain-conserved.” Define the terms clearly up font.

7. Figure captions are very long and have info that could go in methods or results.

8. The references to the figures in the text help immeasurably (e.g. “MA” and “FA”).

Section 2

9. Main point on line 179 is interesting and potentially important. Think about leading the section with that result, since line 169-172 supports that statement. (minor)

Section 3

10. Analysis for statement in line 219?

Section 4

11. Is line 249 the first appearance of the 43 strain-shared sex-based genes with consistent expression? This seems to be relevant right out of the gate (i.e. results section 1). Also related to the clustering question maybe?

Section 5

12. The categories (line 308) are not described in the text. If this is all coming from a previous publication with Diversity Outbred data, it would be nice to describe those results in brief. Fig4C is useful but the reader is forced to stop reading in the middle of a paragraph in order to find that figure and make sense of the results.

13. I also wonder if there isn’t a better way to categorize than LOD scores from eQTL mapping? These are test statistics and dependent on a number of factors, especially the allele frequencies (maybe diplotype frequencies in the DO) and within group variance (measurement error but also genetic background). In any case, if we map an eQTL in male result with confidence of p ≤ 0.1 and map the eQTL in females with confidence of p≤0.0001, I don’t believe that means the eQTL is “stronger” in Females – just that I was lucky enough to map it with more confidence. It also seems that you can compare gene expression levels directly between DO males and females, accounting for eQTL genotype, and avoid this problem.

14. What are “the figures below”? Line 326

15. Is the contention that the local eQTL are in fact correlated with sex-biased CREs even though the casual SNP is not actually in the CRE (and we know this because there is little variation per the previous set of results)?

16. I am not sure I understand the causality here – you propose that the variant + sex controls the CRE which in turn controls gene expression, but there are alternative models that are confounded. This section is a little hard to follow – at the very least, the authors should make sure they do not confuse correlation with causation here. Generally, The results in 323-357 are interspersed with interpretation that may be more appropriate for the discussion section. Certainly, the data are consistent with some of the mechanistic models the authors describe but the data don’t prove causation.

Section 6-8

17. The last four pages of results, along with 5 large figures, consist of specific examples of genes, eQTLs, and their regulation patterns. These are probably useful but distract from the “big picture” results that have been described up to that point. This is likely a matter of preference, but I very much enjoyed the first 4-5 sections of the results and found the last three a slog. The authors may consider condensing these results or moving some parts to supplemental material.

Discussion/Methods

18. The discussion places each result in the context of gene annotations and related literature.

19. The paragraph starting in line 551 is potentially important. Why would the analysis not include all SNPs that segregate between CAST and B6? This undercuts the statements made in the results that there are CREs with no SNPs in them. This is a major concern with the analysis given the importance that it has on the results.

**Have all data underlying the figures and results presented in the manuscript been provided?**

Reviewer #1: Yes

Reviewer #2: Yes

PLOS authors have the option to publish the peer review history of their article (what does this mean?). If published, this will include your full peer review and any attached files.

Reviewer #1: **Yes: **Gary A Churchill

Reviewer #2: No

---

## [Decision Letter · Decision Letter 1]

7 Oct 2021

Dear Dr Waxman,

Thank you very much for submitting your Research Article entitled 'Harnessing natural variation to identify cis regulators of sex-biased gene expression in a multi-strain mouse liver model' to PLOS Genetics.

The manuscript was fully evaluated at the editorial level and by independent peer reviewers. The reviewers appreciated the attention to their concerns and overall were satisfied by the manner in which they were addressed. There was just one remaining point raised by the second reviewer that we believe needs attention. This issue pertains to the assignment of gains and losses within a specific strain. In most cases, it is likely accurate to assume that the ancestral allele matches the allele present in the majority of the founder strains. It is possible, though, that the ancestral allele matches the private allele in the single strain, and that the change occurred on a branch (or independently on multiple branches) that included the majority seven founder strains. This would be highly unlikely for B6, but it is not outside the realm of possibility for CAST. We believe that this issue could be efficiently addressed in the text and would not need additional analyses, but we encourage the authors to do so. This is the only issue that needs to be addressed.

[LINK]

Yours sincerely,

Terry Furey, PhD

Guest Editor

PLOS Genetics

John Greally

Section Editor: Epigenetics

PLOS Genetics

Reviewer's Responses to Questions

**Comments to the Authors:**

Reviewer #1: Thanks for addressing our concerns.

Reviewer #2: This is an interesting set of results and the authors have responded to many of the reviewer concerns. The revised manuscript is easier to read and the text and figures complement and refer to each other better.

I still have a concern with the way eQTL are described as being regulated by one or the other mouse strain. For example – consider the case for which B6 has an "A" and CAST has a "T" nucleotide for a given SNP that is also an eQTL peak. Both reviewers were concerned that the authors were using only a subset of these peaks, but even disregarding that concern there are larger implications for the decision throughout their analysis.

In the scenario above, which is the "regulating strain"? I think the answer is that "regulating strain" cannot be known without the information about the other 6 DO founders and refers to a private SNP among those strains. In other words, a different panel of inbred strains would lead to a different assignment of "regulation." I have been struggling to determine if this is simply a matter of semantics, though there are better terms to use. However, I think the use in the manuscript goes beyond language and has implications for the conclusions.

One moderate concern is that because of the relationships between these founders, we are asked to consider far fewer eQTL "regulated" by B6 than CAST. It is hard to say how that shapes the results because they are sometimes broken out by "regulation" and sometimes lumped together.

A major concern: Does "regulating strain" imply biology? The manuscript suggests that it does – it discusses eQTL function in terms of gain or loss of sex-biased expression. We don't know the evolutionary history of these SNPs, and I am not sure that it is possible to know the state of a common ancestor with the data presented here – especially for the SNPs that are private to CAST. Therefore, it seems misleading to talk about sex-biased expression being gained or lost. Without an ancestral state, the gain or loss is arbitrary – we only know that CAST and B6 are different and that those differences map to a genetic variant. In my toy example above, let's suppose that CAST – the "T" allele – shows sex-biased expression. Neither the statement that B6 lost sex-biased expression nor the statement that CAST gained sex-biased expression can be supported.

While I am generally positive about the data and the analysis, I feel this major concern must be considered.

**Have all data underlying the figures and results presented in the manuscript been provided?**

Reviewer #1: Yes

Reviewer #2: Yes

PLOS authors have the option to publish the peer review history of their article (what does this mean?). If published, this will include your full peer review and any attached files.

Reviewer #1: **Yes: **Gary A Churchill

Reviewer #2: No

---

## [Editor Report · Decision Letter 2]

27 Oct 2021

Dear Dr Waxman,

We are pleased to inform you that your manuscript entitled "Harnessing natural variation to identify cis regulators of sex-biased gene expression in a multi-strain mouse liver model" has been editorially accepted for publication in PLOS Genetics. Congratulations!

Yours sincerely,

Terry Furey, PhD

Guest Editor

PLOS Genetics

John Greally

Section Editor: Epigenetics

PLOS Genetics

Comments from the reviewers (if applicable):

**Data Deposition**

http://datadryad.org/submit?journalID=pgenetics&manu=PGENETICS-D-21-00615R2

**Press Queries**

---

## [Editor Report · Acceptance letter]

3 Nov 2021

PGENETICS-D-21-00615R2 

Harnessing natural variation to identify cis regulators of sex-biased gene expression in a multi-strain mouse liver model 

Dear Dr Waxman, 

We are pleased to inform you that your manuscript entitled "Harnessing natural variation to identify cis regulators of sex-biased gene expression in a multi-strain mouse liver model" has been formally accepted for publication in PLOS Genetics! Your manuscript is now with our production department and you will be notified of the publication date in due course.

With kind regards,

Olena Szabo

PLOS Genetics

On behalf of:
